# Chewing, dental morphology and wear in tapirs (*Tapirus* spp.) and a comparison of free-ranging and captive specimens

**Clemens J. M. Hohl**[1], **Daryl Codron**[2], **Thomas M. Kaiser**[3], **Louise F. Martin**[1], **Dennis W. H. Müller**[4], **Jean-Michel Hatt**[1], **Marcus Clauss**[1]*

1 Clinic for Zoo Animals, Exotic Pets and Wildlife, Vetsuisse Faculty, University of Zurich, Zurich, Switzerland, 2 Department of Zoology and Entomology, University of the Free State, Bloemfontein, South Africa, 3 Center of Natural History, University of Hamburg, Hamburg, Germany, 4 Zoological Garden of Halle, Halle, Germany

* mclauss@vetclinics.uzh.ch

**Data Availability Statement:** All relevant data are within the manuscript and its Supporting Information files.

## Abstract

Feeding practice in herbivorous mammals can impact their dental wear, due to excessive or irregular abrasion. Previous studies indicated that browsing species display more wear when kept in zoos compared to natural habitats. Comparable analyses in tapirs do not exist, as their dental anatomy and chewing kinematics are assumed to prevent the use of macroscopic wear proxies such as mesowear. We aimed at describing tapir chewing, dental anatomy and wear, to develop a system allowing comparison of free-ranging and captive specimens even in the absence of known age. Video analyses suggest that in contrast to other perissodactyls, tapirs have an orthal (and no lateral) chewing movement. Analysing cheek teeth from 74 museum specimens, we quantified dental anatomy, determined the sequence of dental wear along the tooth row, and established several morphometric measures of wear. In doing so, we showcase that tapir maxillary teeth distinctively change their morphology during wear, developing a height differential between less worn buccal and more worn lingual cusps, and that quantitative wear corresponds to the eruption sequence. We demonstrate that mesowear scoring shows a stable signal during initial wear stages but results in a rather high mesowear score compared to other browsing herbivores. Zoo specimens had lesser or equal mesowear scores as specimens from the wild; additionally, for the same level of third molar wear, premolars and other molars of zoo specimens showed similar or less wear compared specimens from the wild. While this might be due to the traditional use of non-roughage diet items in zoo tapirs, these results indicate that in contrast to the situation in other browsers, excessive tooth wear appears to be no relevant concern in *ex situ* tapir management.

## Introduction

Large herbivores are representatives of the landscapes they live in, and diet reconstructions of fossil large herbivores therefore allow reconstructions of past environments. For this purpose,

**Funding:** This study was part of a project funded by the Swiss National Science Foundation (31003A_163300/1 to JMH). DWHM was financed by a candoc Forschungskredit (55220702) by the University of Zurich.

**Competing interests:** The authors have declared that no competing interests exist.

different methods have been developed that link wear signs on teeth, the most frequently preserved element of the vertebrate body, to specific dietary niches [1]. For example, the mesowear method uses the macroscopic appearance of the cusps and the occlusal relief of ungulate cheek teeth to differentiate grazing and browsing herbivores [2].

Tapirs have mostly been excluded from such analyses, because they vary in dental morphology, and their orthal chewing mechanism, from other ungulates (Artiodactyla and Perissodactyla). Tapirs have very brachydont [3], bilophodont teeth (as opposed to the selenodont teeth of most ungulates) [4–7], and use a very limited lateral chewing motion [8, 9]. Thus, even though a tapir molar is depicted in the original report that introduced mesowear [Fig 16 in 2], tapirs were excluded from comparative mesowear databases. For example, Rivals and Lister [10] state that mesowear cannot be applied to tapir teeth. On the other hand, mesowear scoring has recently been applied to fossil tapir species without a proof of concept [11]. One important question in evaluating the applicability of mesowear to a species is whether the signal is stable across several (initial) stages of wear, or whether the mesowear signal varies monotonically with age and wear [12].

Like other perissodactyls, tapir are hindgut fermenters [13]. Tapirs are browsers [14–20]. The fact that they ingest fruits in their natural habitat has, sometimes, led to the assumption that they should be rather considered frugivores than browsers, but various studies have shown consistently that the proportion of fruit in natural diets is typically low, never surpassing 33% in analysed faecal samples [reviewed by 21]. Feeding recommendations are that captive animals should receive a diet based on 70% roughage, mainly lucerne hay, and 30% of a pelleted compound feed [22]. Nevertheless, and even though fundamental differences in nutrient composition between 'wild fruit' and commercially available, domesticated fruit are well known [23–25], tapirs in zoos are often fed diets dominated by commercial fruit, with small proportions of roughage [26].

When comparing mesowear of other browsing ungulates–ruminants and rhinoceroses–between natural habitats and zoos, zoo animals typically show a higher degree of wear, with a signal pointing towards 'grazing' [27–29]. This has been attributed, in those studies, to the putative use of products derived from grasses, either in the form of grass hay, or as grass meal or cereal hulls included in pelleted compound feed. Given the uniformity of that pattern, we would expect a similar difference between free-ranging and zoo tapirs as well, if proxies for wear could be found that facilitate that comparison.

In this contribution, we aimed at first investigating the chewing movements of tapirs in comparison to horses and rhinos, and at describing anatomical measurements of the cheek teeth of three tapir species. For the cheek teeth, we aimed at establishing quantitative measures of dental wear, to develop a score that allowed to ascribe wear stages to individual teeth. We planned to use this wear score to test whether typical mesowear scoring, applied to both the buccal and the lingual side of tapir cheek teeth, showed some consistency across initial wear stages. Finally, we intended to use the quantitative and–if possible–mesowear data of tooth wear to compare free-ranging and zoo specimens, to test whether, as in other browsers, zoo specimens show higher degrees of tooth wear than free-ranging ones.

## Materials and methods

### Mastication

In order to analyze chewing in tapirs, we observed and filmed lowland tapirs of 'Zoo Zürich' in the process of eating hay and a mixture of vegetables (S1 and S2 Videos). Three animals were present during our observations, one adult male, and one adult and one subadult female. We managed to videotape the movement of the mandible during mastication in full frontal view

and produced similar recordings with a domestic horse (*Equus caballus*) (S3 Video) and a greater one-horned rhinoceros (*Rhinoceros unicornis*) (S4 Video) from other facilities for comparison. Further, we used the museum skull of a lowland tapir to reconstruct the possible movement of the mandible, regarding dental occlusion.

## Dentition

**Sample.** The dataset, given *in toto* as S1 Dataset, included 74 individuals of three different tapir species, 51 lowland tapirs (*Tapirus terrestris*), 6 Baird's tapirs (*Tapirus bairdii*) and 17 Malayan tapirs (*Tapirus indicus*). 69 samples were available as dental imprints originating from individual skulls of different European museums, including the Naturhistorisches Museum Basel (NHMB), the National Museums of Scotland Edinburgh (NMS), the Museum für Naturkunde Berlin (ZMB), the Naturmuseum Senckenberg Frankfurt/Main (SMF), the Musée d'Histoire Naturelle Geneva (MHNG), the Zoologisches Museum Hamburg (ZMH), the Natural History Museum London (NHML), the Zoologische Staatssammlung Munich (ZSM), the Staatssammlung für Anthropologie und Paläoanatomie Munich (SAPM), the Musée National d'Histoire Naturelle Paris (MNHN), the Národní Muzeum Prague (NMP), the Naturhistoriska Riksmuseet Stockholm (NRM), the Naturkundemuseum Stuttgart (SMNS), the Naturhistorisches Museum Vienna (NHMW), the Zoologisches Museum Zurich (ZMZ). These dental imprints of either the left or right upper tooth row were available as complete tooth rows. They had been produced according to the methods described in Taylor et al. [29]. On museum specimens, after careful cleaning of the tooth row, a negative mold was taken using sets of two-component polysiloxane dental molding putties (Provil novo Light C. D. 2 fast set EN ISO 4823, type 3, light and Provil novo Putty regular set EN 24823; Heraeus Kulzer GmbH, Hanau, Germany). Positive casts were produced by filling these molds with epoxy resin Injektionsharz EP (Reckli-Chemiewerkstoff, Herne, Germany). Additionally, in 21 specimens, dental casts of either the right or left lower tooth row had been produced. Additionally, museum specimens of 10 complete tapir skulls (8 *T. terrestris*, 2 *T. indicus*) were investigated, measured and documented photographically. For 5 of these complete skulls, dental imprints were also available, resulting in totally 74 individual tapirs, with 26 individuals where measures for both the upper and lower tooth rows were available. The samples comprised free-ranging individuals and individuals from zoos–*T. terrestris* (29 free-ranging/18 zoo/4 unknown), *T. bairdii* (6/0/0), *T. indicus* (6/10/1). Exact ages of the individual specimens were largely unknown. The dental formula of tapir is $I\frac{3}{3}$; $C\frac{1}{1}$; $P\frac{4}{3}$; $M\frac{3}{3}$ with a prominent diastema present between the canines and the first premolars, and a small diastema between $I^3$ and the upper canine, where the lower canine reaches into when the mouth is closed [4, 7] (Fig 1). We use the abbreviations P and M for premolars and molars, respectively, and superscripts (e.g., $M^1$) to indicate maxillary and subscripts (e.g., $M_1$) to indicate mandibular teeth. The eruption sequence of permanent teeth in tapirs was assumed as M1 –(P1-P3)–(P4, M2)–M3 [following 30].

**Quantitative measurements.** Data acquisition was always performed by the same person to reduce bias. A slide caliper was used to measure the teeth. Measurements are listed and explained in Table 1 and illustrated in Fig 2. Tooth length of each tooth was measured in anterioposterior direction on the central axis of a tooth facing the occlusal area. Since tapir teeth are bilophodont, we measured the width of each loph rather than the width of a tooth. Loph width was measured on the base level of each tooth in a buccolingual direction. The mean width of the anterior and posterior loph of one tooth was termed 'tooth width'. Dividing the tooth width of an upper tooth or loph by the width of the occluding lower tooth or loph results in an "index of anisodonty" (ADI) introduced by Fortelius [31], who generally applied this to

A

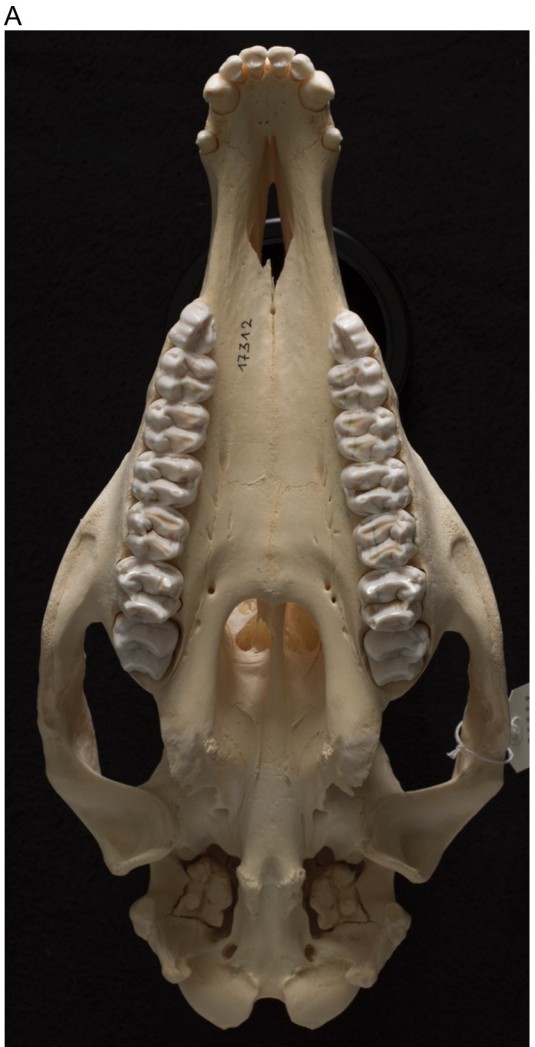

B

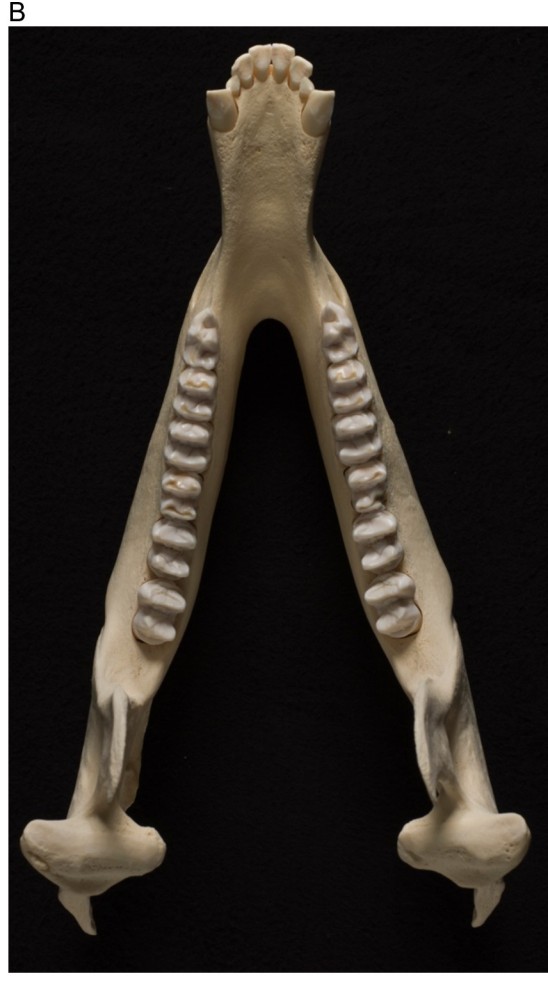

**Fig 1. Tapir teeth.** View of the occlusal surface of the **(A)** maxillary and **(B)** mandibular cheek teeth rows of a lowland tapir (*Tapirus terrestris*).

the M2. In imprints where a boundary between the tooth and its alveolar crest could be distinguished, we measured the distance between the buccal or lingual alveolar crest and the cusp of the associated loph for all four cusps when possible ('cusp height'), calculating 'Δ cusp height' by subtracting the lingual cusp height from the buccal cusp height of the same loph. We measured the minimum distance (in anterioposterior direction) between the crests of the (anterior) metaloph and the (posterior) protoloph, if the enamel was still intact. If a dentine basin was present, this value was measured as the minimum distance between the posterior enamel ridge of the anterior loph and the anterior enamel ridge of the posterior loph. When dentine basins of two lophs of a tooth are connected, this value is zero–except if the connection is only present on the buccal part of the tooth, as we always measured this value on the lingual part. Defining wear stages by the presence and dimension of a dentine basin, we used this minimum loph distance divided by tooth length of the same tooth as 'relative loph distance' (RLD), which becomes shorter with increasingly worn teeth. We measured the maximum distance between two enamel ridges of a dentine basin in anterioposterior direction ('dentine basin

**Table 1. Quantitative measures of the present study.** See Fig 2 for illustration.

| Measure | | Description | Change with wear |
|---|---|---|---|
| Tooth length | | Anterioposterior dimension on the central axis of a tooth on the occlusal area | no change |
| Loph distance | | On lingual part of the tooth: minimum distance (in anterioposterior direction) between the crests of the (anterior) metaloph and the (posterior) protoloph, if enamel still intact; if dentine basin present, minimum distance between the posterior enamel ridge of the anterior loph and the anterior enamel ridge of the posterior loph; when dentine basins of two lophs of a tooth are connected, this value is zero, except if the connection is only present on the buccal part of the tooth | decrease |
| Basin length | anterior and posterior | Maximum distance between two enamel ridges of a dentine basin in anterioposterior direction | increase |
| Loph width | anterior and posterior | Buccolingual dimension on the base level of each loph | no change |
| Tooth width | | Average of anterior and posterior loph width | no change |
| Basin width | anterior and posterior | Maximum distance between two enamel ridges of a dentine basin in buccolingual direction | increase |
| Cusp height | buccal and lingual, anterior and posterior | Distance between the buccal or lingual alveolar crest and the cusp of the associated loph | change |
| Anisodonty index | ADI | Width of maxillary tooth or loph / Width of mandibular tooth or loph | no change |
| Relative loph distance | RLD | Loph distance / Tooth length | decrease |
| Relative loph length | RLL | Basin length / Tooth length | increase |
| added RLL | aRLL | (Anterior + Posterior basin length) / Tooth length | increase |
| Relative loph width | RLW | Basin width / Loph width | increase |
| added RLW | aRLW | (Anterior + Posterior basin width) / (Anterior + Posterior loph width) | increase |
| Δ cusp height | | Cusp height buccal–Cusp height lingual | increase |

length') and buccolingual direction ('dentine basin width') to define dimension of dentine basins. The 'relative loph length' (RLL) is calculated by dividing a loph's dentine basin length by the length of its tooth; the 'added relative loph length' (aRLL) is the sum of the dentine basin lengths of two lophs of a tooth and divided by the tooth length. This value represents a whole tooth, rather than one loph, and is in theory the reciprocal value to RLD except for the enamel ridges. The relative loph width (RLW) is the dentine basin width divided by loph

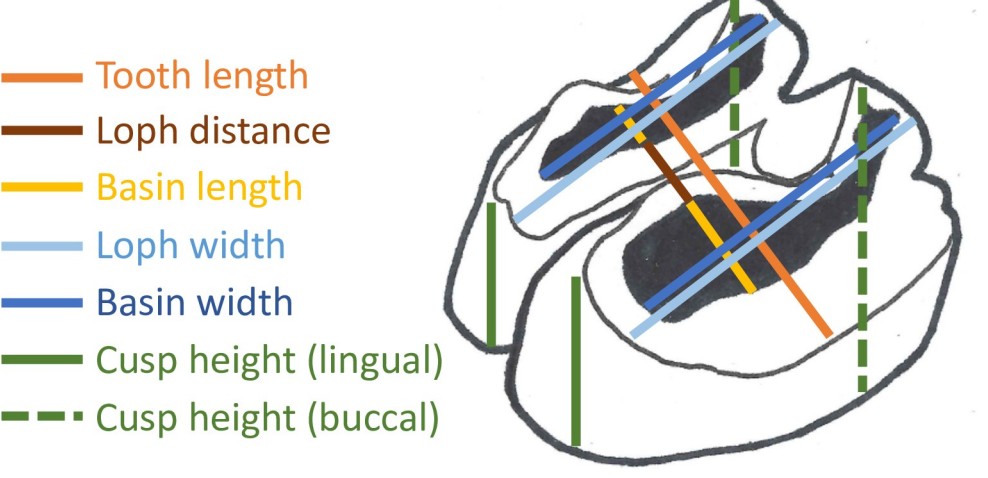

**Fig 2. Measurements.** Linear measurements taken in the present study.

width, and the added relative loph width (aRLW) is the sum of the dentine basin widths of the anterior and posterior loph divided by the sum of the width of these two lophs. Epoxy casts of the lower tooth row were measured in the same manner as described for upper tooth row, except for cusp height. Note that the first premolar is absent in the lower tooth row.

**Scores.** To quantify abrasion, we developed a macrowear scoring system (see Results). Macrowear scoring systems, based on verbal descriptions (from no wear to slight/little wear, moderate wear, to heavy wear) for tapirs have developed before [32–34]; we aimed at providing anatomy-based descriptions for these stages. Since dental morphology between different tapir species is comparable, the scoring system does not vary between them. The macrowear score is based on the presence and morphology of dentine basins. Due to the triangular shape of the first premolar, this tooth is not comparable to the other cheek teeth and is therefore not considered in our scoring system. Regarding wear stages, we limited ourselves to the upper tooth rows. The scoring system includes five wear stages from little to no wear (1) up to highly worn teeth with a concave occlusal surface (5). The scoring system is designed to classify individual lophs; when comparing wear of whole teeth, the average macrowear score of two lophs of a tooth is used. We applied our scoring system on all teeth that were available as epoxy casts (n = 69).

Finally, we scored the mesowear of our samples as described in Taylor et al. [35], where we used the extended mesowear scale (EM). Note that mesowear is typically scored on the ectoloph, which does not exist in tapirs as such. The original mesowear method was explicitly not intended for dental morphologies lacking an ectoloph [2, 36], irrespective of its later applications [reviewed in 37] that also included tapirs [11]. We employed the principles of mesowear scoring to the buccal and lingual side of the cheek teeth. The cusp shape is classified into 5 groups: 'sharp' = sharp cusp with naked eye and with a magnifying glass; 'round-sharp' = sharp cusp with naked eye, but round with magnifying glass; 'round' = clearly round, cusp curvature length < ½ of the cusp length; 'round-round' = clearly round, cusp curvature length > ½ of the cusp length; 'blunt' = highest point of cusp not clear. Occlusal relief is also categorized into 5 groups: 'high-high' = angle of the valley between cusps is less than 90 degrees; 'high' = angle of the valley between cusps is more than 90 degrees, but the height of the valley (x) divided by tooth length (y) is between 0.125 and 0.25; 'high-low' = x/y is between 0.05 and 0.125; 'low' = x/y is between 0 and 0.05; 'flat-negative' = highest point of cusp not clear, x/y ~ 0. We applied the scoring scheme to the lingual (hypocone and protocone) and buccal cusps (metacone and paracone) of the upper teeth. Regarding occlusal relief, we also scored both, the lingual and buccal side. Mesowear scores were transcribed into numerical data considering 's' for cusp shape and 'hh' for occlusal relief as the lowest score (1) and 'b' for cusp shape and 'fn' for occlusal relief as the highest score (5). To compare our mesowear scores with data from Kaiser et al. [38], we used the approach combining cusp shape and occlusal relief into a single data as described in Kaiser et al. [28]. For this purpose, we transcribed our mesowear score to the required original scheme: for cusp score, 's' and 'rs' were considered sharp, 'r' was considered round and 'rr' and 'b' were considered blunt, while for the occlusal relief, 'hh' and 'h' were considered high and 'hl', 'l' and 'fn' were considered low. The tapir species' hypsodonty index and categorical habitat variable were taken from Mendoza and Palmqvist [3].

**Statistics.** Statistics were performed using R v 3.5.2 [39], with the lmerTest package [40]. We mainly used linear correlation analysis (primarily Pearson's *R* for normally-distributed data, otherwise we used Spearman's ρ) to assess relationships between measurements. The resulting statistics are given in the corresponding figure legends. These tests were made for each species separately. To compare mesowear scores between free-ranging and zoo individuals, we used generalized linear models incorporating the effect 'origin' (natural habitat or zoo), assuming poisson distributions of mesowear variables. These models were repeated for

each species and tooth separately. As an additional test for differences between free-ranging and zoo individuals, we used general linear models (confirming normal distribution of residuals) with the quantitative wear measure (relative loph distance RLD) of the $M^3$ as the independent variable, the RLD of the tooth in question as the dependent variable, and the indviduals' status (free-ranging or zoo) as cofactor. First, we also included the RLD $M^3$ x status interaction in all models (to test whether there were different slopes in specimens from natural habitats or zoos); as these interactions were always not significant, we report only the results of models without the interactions here. For both sets of models, equality of residual variances was confirmed using Levene's tests, and, in the case of general linear models, normality of residuals was confirmed using Shapiro-Wilk's tests. In the one instance of a non-normal distribution of residuals (*T. terrestris* P4), rerunning the model using ranked data (i.e. a non-parametric version) yielded qualitatively similar results. The significance level was set to 0.05 throughout.

## Results

### Mastication

When observing masticatory movement of the tapirs, there was no obvious lateral movement of the mandible. The power stroke seemed to be orthal without any grinding movement or excursion of the mandible. The cheeks bulged out more or less symmetrically during the power stroke (Fig 3). When viewed laterally so that only one cheek was visible, this bulging created the impression of a lateral chewing stroke. In comparison, the domestic horse and the greater one-horned rhinoceros displayed a remarkable lateral movement of the lower jaw during mastication, which is easily recordable (Fig 3). The tapirs were observed while feeding on a mixture of vegetables, which are not the main component of their diet. We attempted to produce similar recordings of tapirs feeding on forage (lucerne hay), but due to their feeding behavior, forage always stood out of their mouths, obscuring jaw and cheek movements and making a visual evaluation of movement impossible.

The upper third incisors and lower canines on a tapir skull are very prominent (Fig 4). When attempting to manually execute a grinding movement on the tapir skull, the third upper incisor was blocked by the lower canine and therefore inhibited lateral movement of the mandible. When opening the jaw as far as needed to excurse the mandible laterally by more than a few millimeters, the cheek teeth row was no longer in complete occlusion. By contrast, on several horse skulls, a similar simulation of lateral movements was possible while maintaining occlusion of the complete cheek tooth row.

### Dental anatomy

The maxillary tooth row was longer than the mandibular one in all species, which is linked to the absence of the $P_1$ (Fig 5). In contrast to the generally quadrangular shape of the cheek teeth, the $P^1$ has a triangular shape, and therefore only a single loph (Fig 1). The $M^2$ was the widest tooth of all cheek teeth, in particular its anterior loph (Fig 5). In the mandibular tooth row, the anterior loph of the $M_3$ was the widest structure. Maxillary teeth were always wider than mandibular teeth (Fig 5, Table 2), leading to an anisodonty index of 1.26–1.28 in the three tapir species. At rest, the lingual side of the maxillary and mandibular teeth were in alignment, and therefore, the maxillary teeth extended buccally over the mandibulary ones (Fig 4).

In maxillary teeth, the anterior loph had a lesser width than the posterior loph for the $P^2$, both lophs were similar in width for $P^3$, and the anterior loph had a greater width than the posterior loph for all other teeth, with the largest difference in the $M^3$ (Fig 6). For the mandibular teeth, premolars generally had wider posterior, and molars wider anterior lophs (Fig 6).

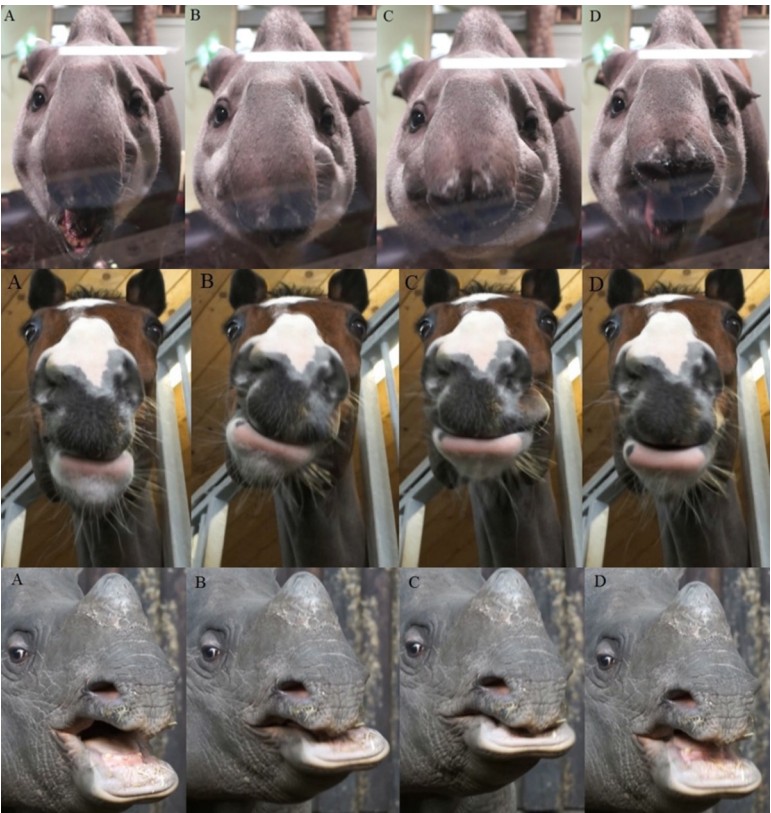

**Fig 3. Chewing patterns.** Video screenshots of a lowland tapir (*Tapirus terrestris*), domestic horse (*Equus caballus*) and greater one-horned rhinoceros (*Rhinoceros unicornis*) during mastication. **A**: Mouth in full extension; **B**: Beginning of power stroke; **C**: End of power stroke; **D**: Mouth opening again.

When relating tooth length to loph width in the maxillary row, the $P^1$ was longer than wide, whereas all other teeth were generally wider than long, with the $P^2$ being most extreme and the $M^2$ closest to a quadratic shape (Fig 7).

The shape change was mainly related to the posterior loph: whereas the relationship of the anterior loph width and tooth length was more or less constant (Fig 8), the relationship of the posterior loph width and tooth length indicated that absolute posterior loph width was constant across tooth lengths from $P^2$ to $M^3$ (Fig 9).

For the mandibular row, these patterns were similar, but mandibular teeth were generally longer than wide, and the difference between the anterior and posterior lophs were not as distinct as for the maxillary row (Figs 7–9).

## Quantitative wear measures

The relative loph distance (RLD) in teeth with an intact enamel crest in both lophs ('unworn' teeth) ranged from 31.2% to 56.6%, depending on tooth position and species. For those individuals where both maxillary and mandibular teeth could be measured, there was a consistent correlation between maxillary and mandibular RLD (Fig 10). Across the whole sample, the average RLD of teeth correlated with their eruption sequence, suggesting that older teeth experienced, on average, more wear (Fig 11).

By definition, the relative loph length (RLL) in unworn teeth is zero, because the loph is represented by a single ridge and not by a dentine basin. The correlation of the anterior and posterior RLL of individual teeth indicated a generally even wear, which is more advanced on the

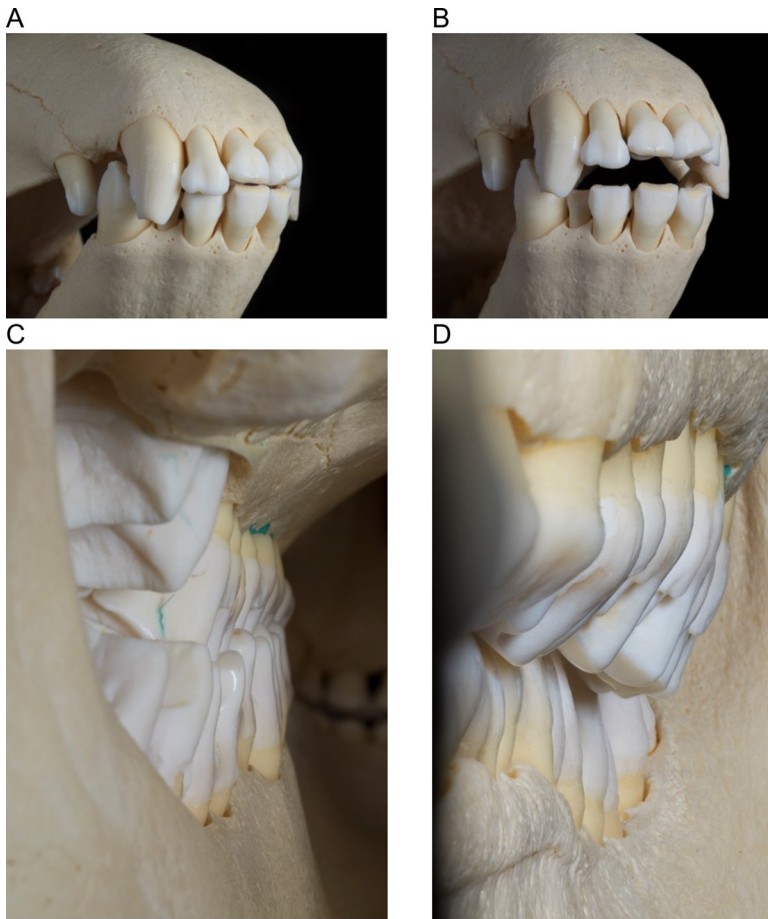

**Fig 4. Detailed tapir dentition.** Craniolateral view onto the incisors and canines of a lowland tapir (*T. terrestris*) in (**A**) closed and (**B**) opened position. View of the cheek tooth row of the same skull in occlusion (**C**) viewed from caudal to rostral along the lingual side of the left cheek tooth row, indicating little overlap on the lingual side, and (**D**) viewed from rostral to caudal along the buccal side of the left cheek tooth row, showing overlap of the maxillary teeth on the buccal side.

posterior loph of P2 and often more advanced on the anterior loph of the upper molars (Fig 12). For those individuals where both maxillary and mandibular teeth could be measured, there was a consistent correlation between maxillary and mandibular RLL (Fig 10) except for the P2, which showed an irregular pattern. Across the whole sample, the average aRLL of teeth correlated with their eruption sequence, again suggesting on average more wear on older teeth (Fig 11). There was a strict correlation between the RLD and the aRLL (Fig 13).

The relative loph width (RLW) is also, by definition, zero for unworn teeth, as there is no dentine basin. Again, the correlation of the anterior and the posterior RLW of individual teeth showed that wear of P2 is more advanced on the posterior loph, and that for molars, wear is typically more advanced on the anterior lophs until the posterior ones catch up at later wear stages (Fig 12). For those individuals where both maxillary and mandibular teeth could be measured, there was no evident correlation between maxillary and mandibular RLW (Fig 10). Across the whole sample, the average aRLW of teeth correlated with their eruption sequence, again suggesting on average more wear on older teeth (Fig 11). There was a strict correlation between the RLD and the aRLW (Fig 13), and between aRLL and the aRLW, when excluding P$^2$ (Fig 13).

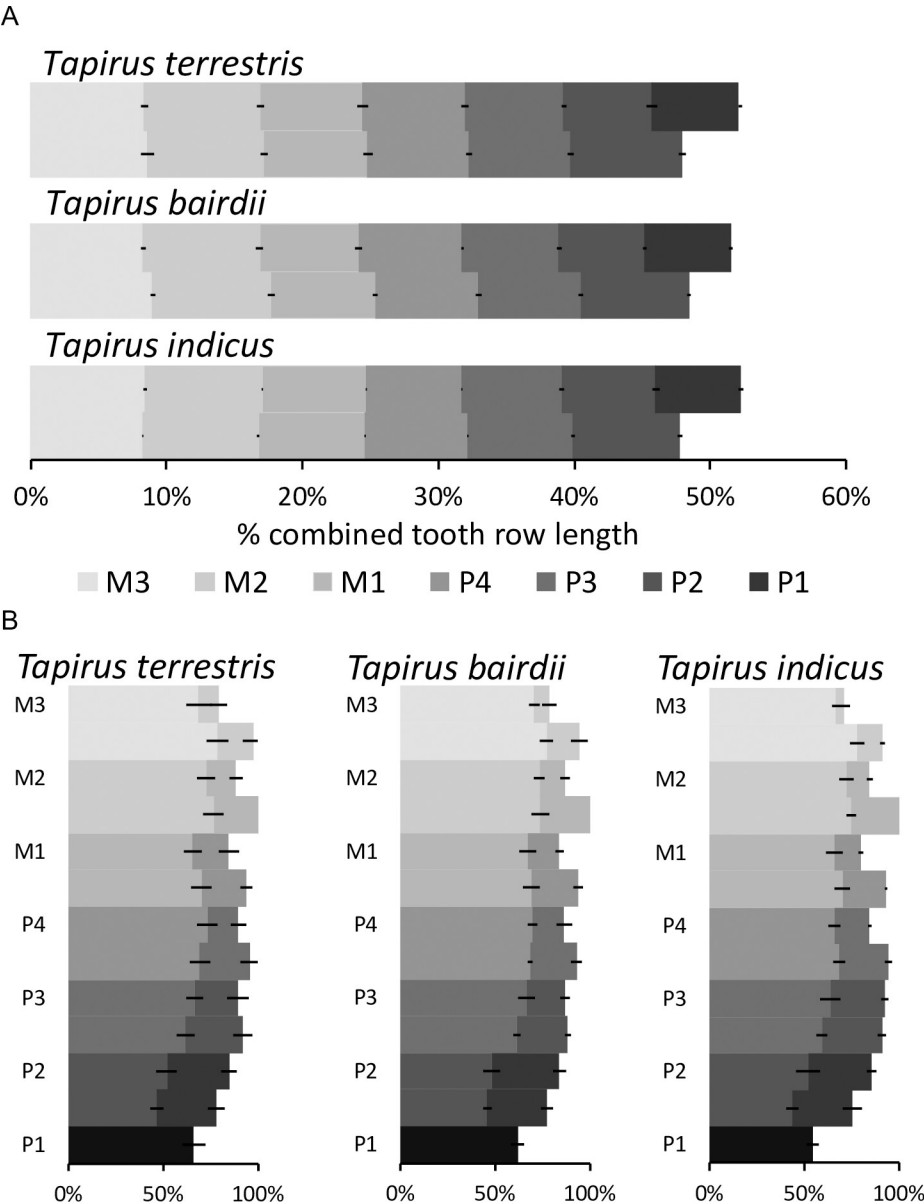

**Fig 5. Tapir tooth dimensions.** Mean (± SD) **(A)** length of individual teeth of the maxillary and mandibular cheek tooth row in relation to the total length of both rows combined (100% = length of maxillary + mandibular cheek tooth row). *T. terrestris* (n = 13), *T. bairdii* (n = 4), *T. indicus* (n = 2); **(B)** relative width of the anterior and posterior lophs of individual teeth in relation to the width of the anterior loph of the $M^2$ (which is set at 100%). The maxillary tooth row is illustrated with a darker shade, compared to the mandibular, overlying tooth row. There is no $P_1$. $P^1$ only has one loph. Note that in all cases, maxillary teeth are wider than mandibular ones. *T. terrestris* (n = 17), *T. bairdii* (n = 6), *T. indicus* (n = 3).

Regarding mandibular cheek teeth, correlation between RLD and aRLL is present and linear (Fig 14). aRLW correlates with RLD and aRLL, but in a fashion that indicates that whereas the length-oriented wear measures (RLD, aRLL) are continuously indicating progressive wear, the width-oriented wear measure (aRLW) soon reaches its maximum and is constant from then on (Fig 14).

**Table 2. Tapir teeth measurements.** Average loph width (in mm) (al = anterior loph; pl = posterior loph), tooth width (mm) and Anisodonty Index (ADI) per loph and tooth.

| | | P2 al | P2 pl | P3 al | P3 pl | P4 al | P4 pl | M1 al | M1 pl | M2 al | M2 pl | M3 al | M3 pl |
|---|---|---|---|---|---|---|---|---|---|---|---|---|---|
| T. terrestris (n = 17) | Upper jaw | 18.11 ± 0.94 | 19.66 ± 0.94 | 21.38 ± 1.30 | 20.77 ± 1.46 | 22.19 ± 1.41 | 20.61 ± 1.33 | 21.77 ± 1.14 | 19.49 ± 1.33 | 23.10 ± 1.24 | 20.39 ± 1.07 | 22.62 ± 1.46 | 18.44 ± 1.07 |
| | | 18.89 ± 0.83 | | 21.07 ± 1.35 | | 21.40 ± 1.29 | | 20.63 ± 1.14 | | 21.75 ± 1.06 | | 20.63 ± 1.14 | |
| | Lower jaw | 10.77 ± 0.63 | 11.89 ± 0.85 | 14.22 ± 0.79 | 15.30 ± 0.83 | 15.95 ± 0.89 | 16.89 ± 0.83 | 16.17 ± 1.02 | 15.09 ± 0,94 | 17.62 ± 0.78 | 16.73 ± 0,72 | 18.10 ± 0,81 | 15.88 ± 1.19 |
| | | 11.32 ± 0.57 | | 14.76 ± 0.76 | | 16.42 ± 0.81 | | 15.63 ± 0.96 | | 17.18 ± 0.72 | | 16.90 ± 0.93 | |
| | ADI | 1.68 ± 0.10 | 1.66 ± 0.13 | 1.51 ± 0.11 | 1.36 ± 0.11 | 1.39 ± 0.11 | 1.22 ± 0.09 | 1.35 ± 0.09 | 1.29 ± 0.09 | 1.31 ± 0.09 | 1.22 ± 0.09 | 1.25 ± 0.08 | 1.18 ± 0.08 |
| | | **1.68 ± 0.09** | | **1.43 ± 0.10** | | **1.31 ± 0.10** | | **1.32 ± 0.08** | | **1.27 ± 0.08** | | **1.22 ± 0.06** | |
| T. bairdii (n = 6) | Upper jaw | 18.24 ± 0.81 | 19.80 ± 1.11 | 20.87 ± 0.97 | 20.50 ± 1.18 | 21.93 ± 1.26 | 20.40 ± 1.07 | 22.08 ± 0.70 | 19.76 ± 1.00 | 23.62 ± 1.08 | 20.55 ± 1.02 | 22.56 ± 1.08 | 18.76 ± 0.48 |
| | | 19.02 ± 0.84 | | 20.68 ± 1.07 | | 21.17 ± 1.14 | | 20.92 ± 0.81 | | 22.08 ± 1.00 | | 20.66 ± 0.54 | |
| | Lower jaw | 10.98 ± 0.75 | 11.42 ± 0.65 | 14.75 ± 0.64 | 15.69 ± 1.59 | 16.14 ± 0.62 | 16.49 ± 0.83 | 16.25 ± 0.74 | 15.85 ± 1.02 | 17.42 ± 0.39 | 17.28 ± 0.65 | 18.00 ± 0.88 | 16.51 ± 1.29 |
| | | 11.20 ± 0.32 | | 15.52 ± 1.10 | | 16.31 ± 0.68 | | 15.92 ± 0,82 | | 17.26 ± 0.37 | | 17.63 ± 0.81 | |
| | ADI | 1.69 ± 0.14 | 1.74 ± 0.15 | 1.45 ± 0.02 | 1.32 ± 0.06 | 1.36 ± 0.04 | 1.24 ± 0.06 | 1.36 ± 0.05 | 1.25 ± 0.08 | 1.36 ± 0.09 | 1.19 ± 0.04 | 1.25 ± 0.04 | 1.11 ± 0.08 |
| | | **1.73 ± 0.06** | | **1.37 ± 0.03** | | **1.30 ± 0.05** | | **1.32 ± 0.06** | | **1.28 ± 0.06** | | **1.18 ± 0.04** | |
| T. indicus (n = 3) | Upper jaw | 19.67 ± 0.85 | 22.34 ± 0.95 | 23.81 ± 1.53 | 24.15 ± 0.88 | 25.09 ± 1.01 | 22.49 ± 1.15 | 24.36 ± 1.18 | 20.84 ± 1.45 | 26.16 ± 1.45 | 22.03 ± 0.84 | 24.22 ± 1.15 | 18.90 ± 0.39 |
| | | 21.00 ± 0.47 | | 23.98 ± 1.17 | | 23.79 ± 1.08 | | 22.60 ± 1.31 | | 24.09 ± 1.14 | | 21.56 ± 0.77 | |
| | Lower jaw | 11.59 ± 0.08 | 13.44 ± 1.13 | 15.50 ± 0.60 | 16.63 ± 1.15 | 18.19 ± 0.28 | 17.45 ± 0.16 | 18.28 ± 0.07 | 17.14 ± 0.43 | 19.50 ± 0.88 | 18.90 ± 0.60 | 20.68 ± 0.29 | 17.72 ± 0.56 |
| | | 12.12 ± 0.08 | | 15.60 ± 0,86 | | 17.82 ± 0.22 | | 17.57 ± 0.20 | | 18.99 ± 0.71 | | 19.20 ± 0.42 | |
| | ADI | 1.65 ± 0.05 | 1.68 ± 0.20 | 1.54 ± 0.11 | 1.46 ± 0.10 | 1.38 ± 0.03 | 1.29 ± 0.05 | 1.33 ± 0.07 | 1.22 ± 0.10 | 1.34 ± 0.04 | 1.17 ± 0.04 | 1.17 ± 0.04 | 1.07 ± 0.01 |
| | | **1.73 ± 0.07** | | **1.50 ± 0.10** | | **1.33 ± 0.04** | | **1.28 ± 0.08** | | **1.26 ± 0.04** | | **1.12 ± 0.02** | |

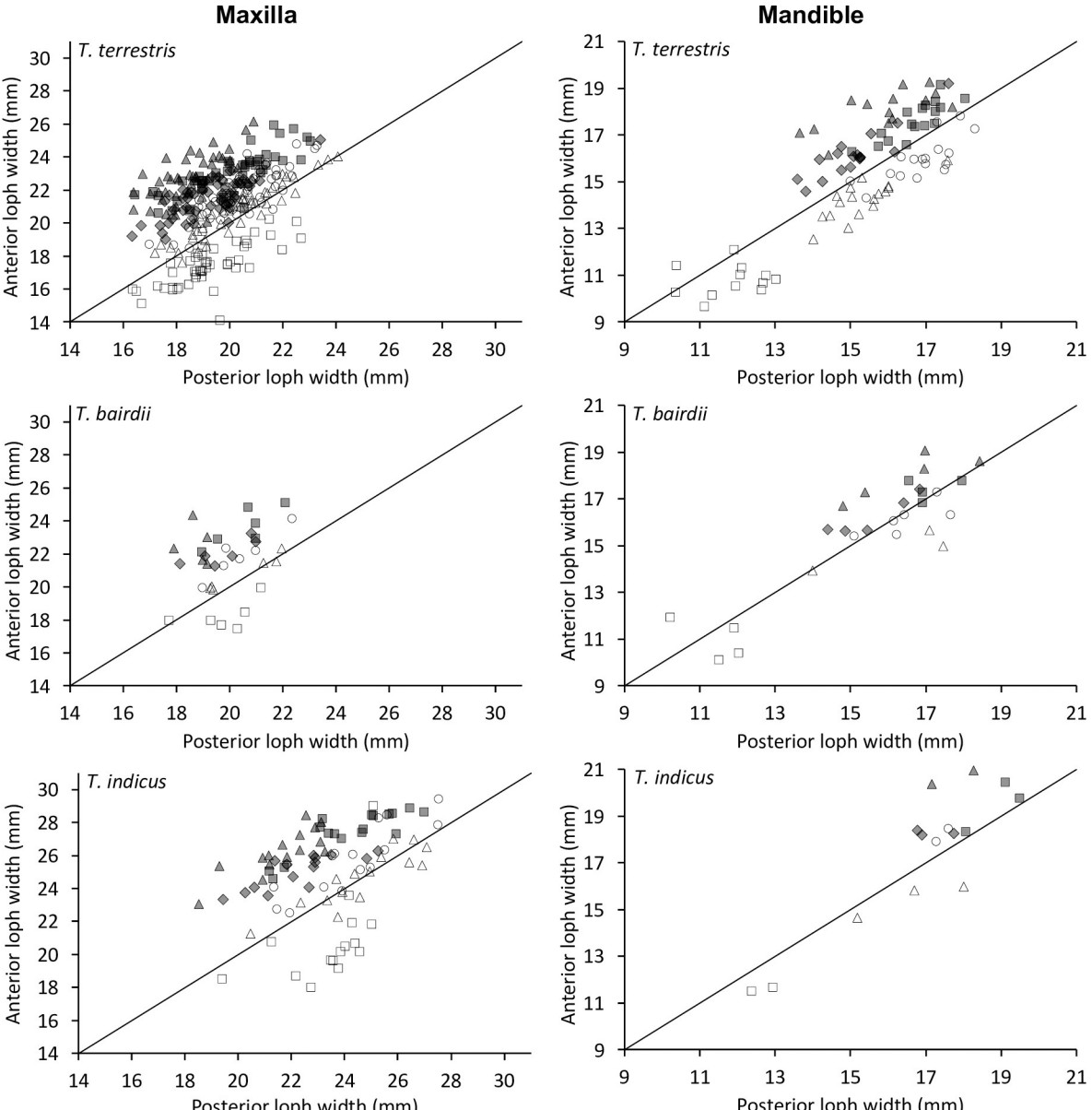

**Fig 6. Tapir teeth–loph dimensions.** Comparison of posterior loph width and the anterior loph width of the upper cheek teeth in three tapir species. White rectangle = P2; White triangle = P3; White circle = P4; Grey diamond = M1; Grey rectangle = M2; Grey triangle = M3. Pearson's correlations: *T. terrestris*: maxilla n = 295, *R* = 0.45, *P*<0.001; mandible n = 94, *R* = 0.81, *P*<0.001. *T. bairdii*: maxilla n = 35, *R* = 0.34, *P* = 0.043; mandible n = 27, *R* = 0.87, *P*<0.001. *T. indicus*: maxilla n = 93, *R* = 0.38, *P*<0.001; mandible n = 15, *R* = 0.90 *P*<0.001. The line denotes y = x.

In unworn teeth, the cusp height was higher on the buccal than on the lingual side in $P^2$, and lower in $P^4$, $M^2$ and $M^3$ (there were no unworn $P^3$ and $M^1$). In other words, the Δ cusp height (buccal minus lingual) was negative for the main maxillary cheek teeth when unworn. This changed systematically with wear: there was a negative correlation of Δ cusp height with RLD for all maxillary teeth (Fig 15). In other words, as teeth were worn, the lingual height first became similar to, and then fell below, buccal height. This led to the visual impression observed in many casts that the buccal side of the maxillary tooth row protrudes like a rail (Fig

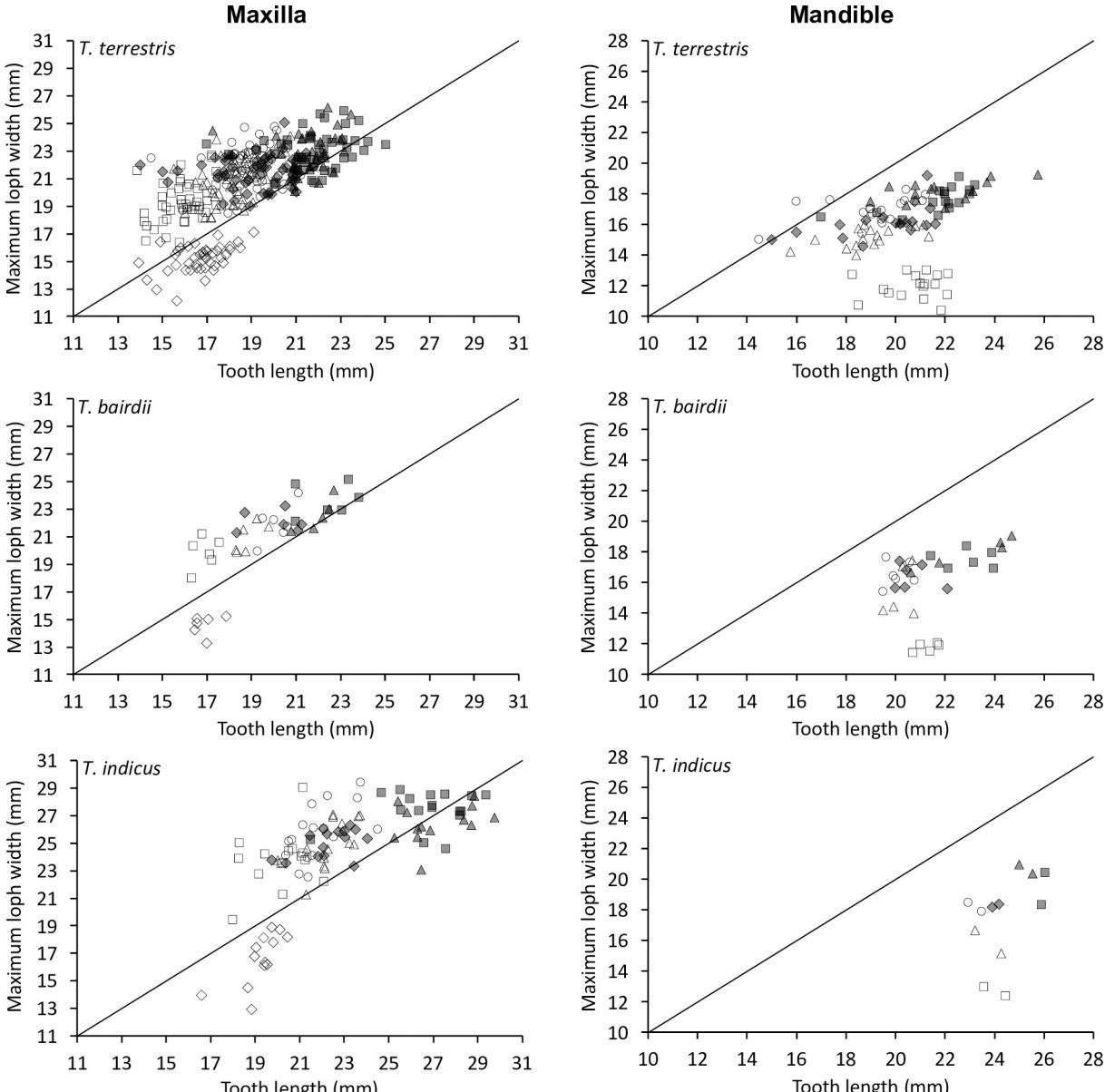

**Fig 7. Tapir teeth–length and width.** Comparison of tooth length and maximum loph width in three tapir species. White diamond = P1; White rectangle = P2; White triangle = P3; White circle = P4; Grey diamond = M1; Grey rectangle = M2; Grey triangle = M3. Pearson's correlations: *T. terrestris*: maxilla n = 291, *R* = 0.53, *P*<0.001; mandible n = 94, *R* = 0.21, *P* = 0.042. *T. bairdii*: maxilla n = 35, *R* = 0.66, *P*<0.001; mandible n = 27, *R* = 0.27, *P* = 0.170. *T. indicus*: maxilla n = 91, *R* = 0.47, *P*<0.001; mandible n = 12, *R* = 0.41 *P* = 0.181. The line denotes y = x.

16). Height measurements were not performed on mandibular cheek teeth, because wear seemed to be evenly on both cusps when observing different tapir dentitions.

## Wear scores

Based on the state of the dentine basins, a macrowear score was developed that is applied on the basis of individual teeth (Fig 17). The macrowear score showed a high correlation with the RLD, aRLL and aRLW (Fig 18).

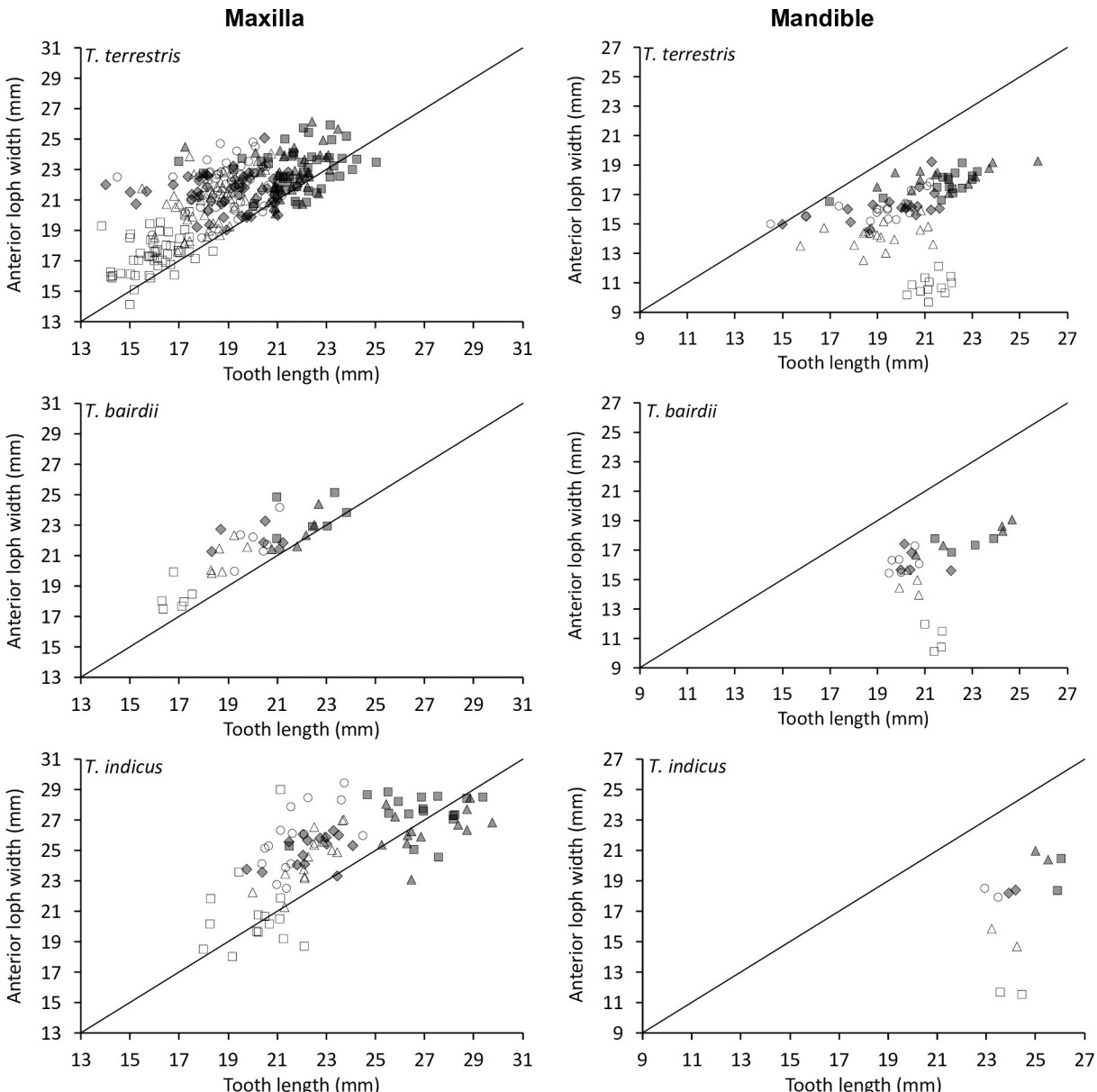

**Fig 8. Tapir teeth–shape.** Comparison of tooth length and anterior loph width in three tapir species. White rectangle = P2; White triangle = P3; White circle = P4; Grey diamond = M1; Grey rectangle = M2; Grey triangle = M3. Pearson's correlations: *T. terrestris*: maxilla n = 334, *R* = 0.71, *P*<0.001; mandible n = 94, *R* = 0.27, *P* = 0.008. *T. bairdii*: maxilla n = 41, *R* = 0.83, *P*<0.001; mandible n = 28, *R* = 0.34, *P* = 0.080. *T. indicus*: maxilla n = 104, *R* = 0.73, *P*<0.001; mandible n = 12, *R* = 0.44 *P* = 0.157. The line denotes y = x.

The mesowear scores for cusp shape and occlusal relief correlated mainly for the lingual side, while on the buccal side, there was less correlation, indicating a more stable pattern (Fig 19). Comparing cusp shape and occlusal relief individually to the macrowear score, there were positive correlations for both on the lingual side, whereas on the buccal side, there were no correlations, indicating that the mesowear signals were constant especially across the early wear stages (Fig 20). The calculated mesowear score of the $M^2$ according to Kaiser et al. [28] for free-ranging tapirs of the macrowear score range 1–3 were: *T. terrestris* = 1.17 (± 1.12, n = 29); *T. bairdii* = 2.17 (± 1.07, n = 6); *T. indicus* = 0.20 (± 0.40, n = 5).

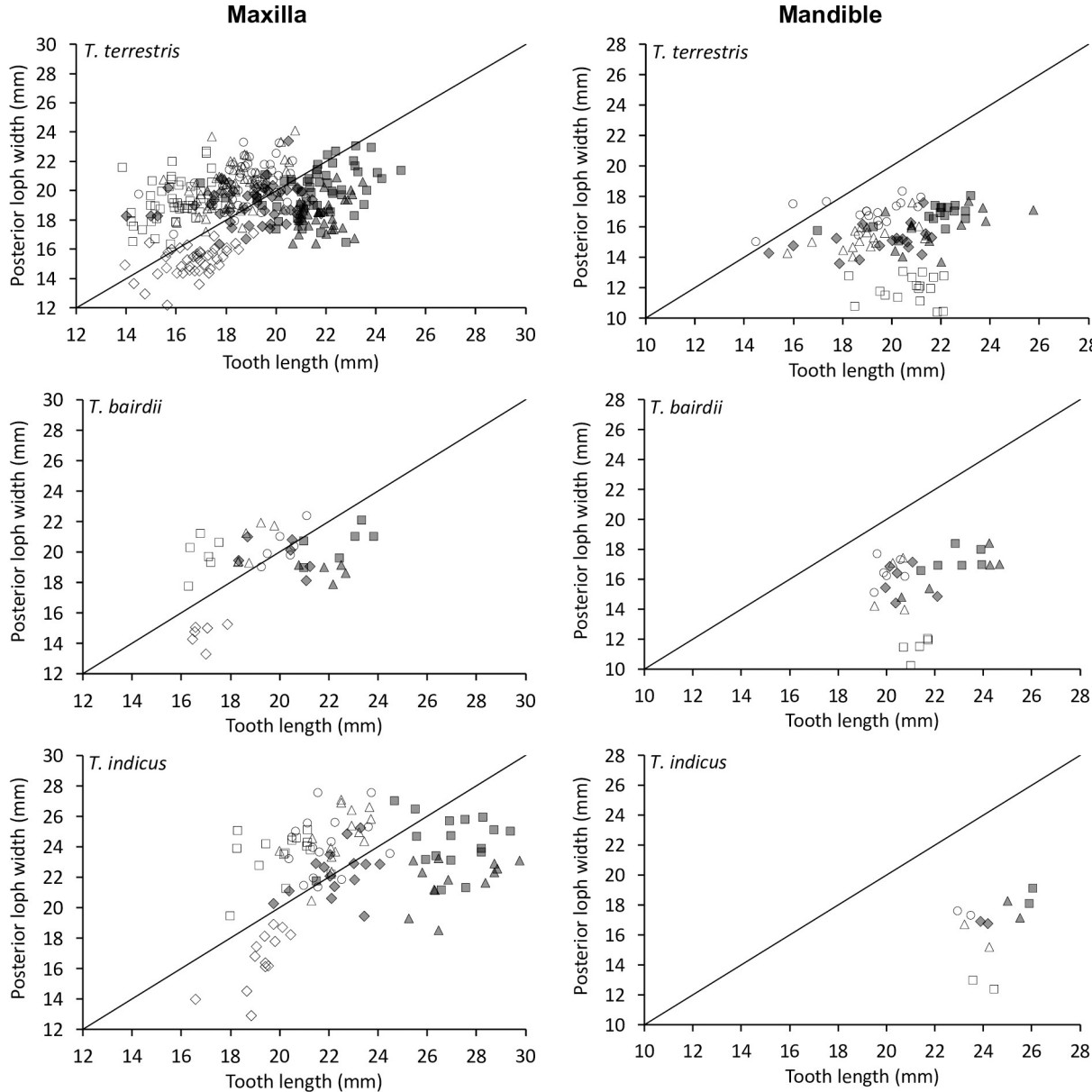

**Fig 9. Tapir teeth–shape.** Comparison of tooth length and posterior loph width in three tapir species. White diamond = P1; White rectangle = P2; White triangle = P3; White circle = P4; Grey diamond = M1; Grey rectangle = M2; Grey triangle = M3. Pearson's correlations: *T. terrestris*: maxilla n = 291, $R = 0.10$, $P = 0.100$; mandible n = 99, $R = 0.14$, $P = 0.182$. *T. bairdii*: maxilla n = 35, $R = 0.05$, $P = 0.764$; mandible n = 32, $R = 0.28$, $P = 0.123$. *T. indicus*: maxilla n = 91, $R = 0.01$, $P = 0.931$; mandible n = 12, $R = 0.37$ $P = 0.243$. The line denotes y = x.

## Comparisons natural habitat–zoo

When comparing buccal mesowear scores between animals from natural habitats and zoos (using only animals with macrowear scores 1–3 to exclude heavily worn specimens), there were no significant differences for *T. indicus* (Table 3). For *T. terrestris*, both cusp shape and occlusal relief of the $M^1$ indicated significantly more wear in animals from natural habitats; additionally, trends of a similar direction appeared for the occlusal relief of $P^3$ ($P = 0.060$), and the cusp shape ($P = 0.056$) and occlusal relief ($P = 0.081$) of $P^4$ (Table 3).

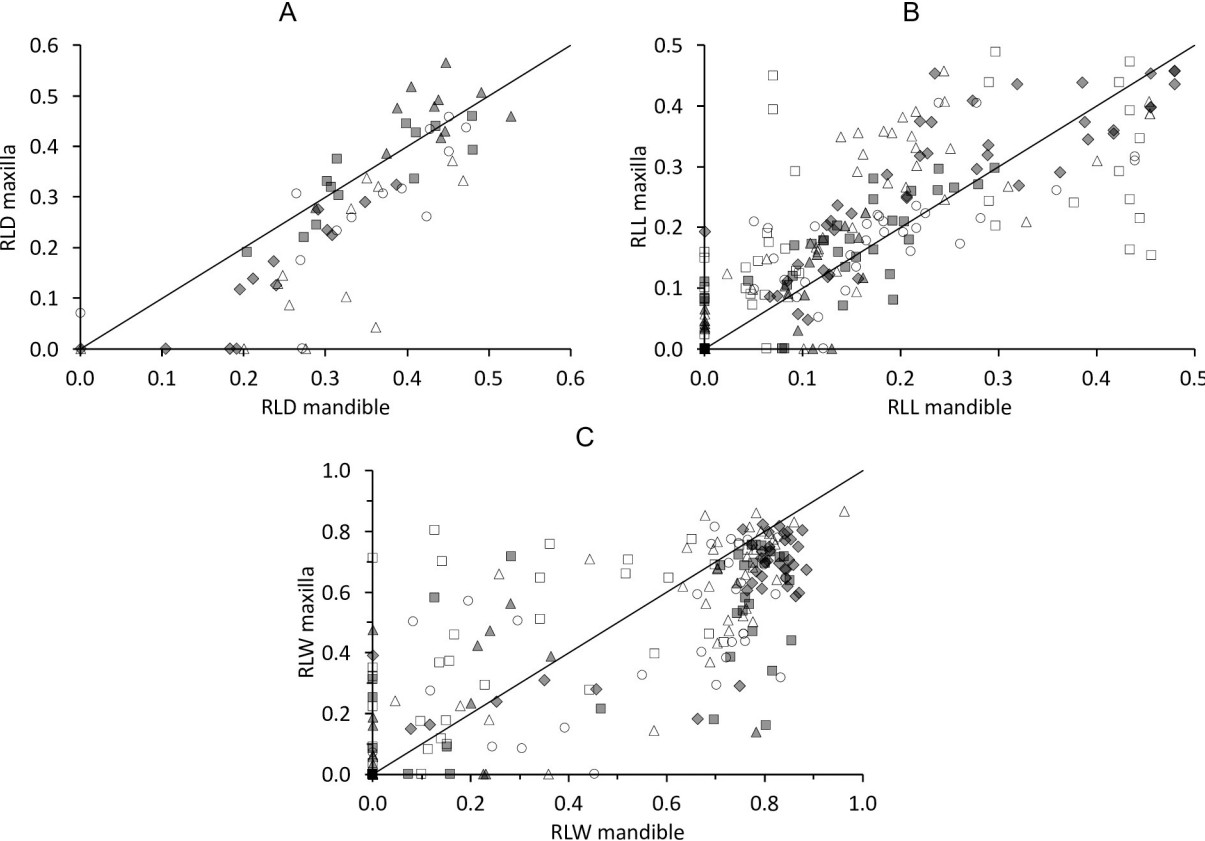

**Fig 10. Tapir tooth wear–wear upper vs lower cheek teeth.** Comparisons of the maxillary and mandibular teeth of three tapir species for the quantitative wear measuremens (**A**) relative loph distance (RLD) for individual teeth, (**B**) relative loph length (RLL) for individual lophs, (**C**) relative loph width for individual lophs. White triangle = P3; White circle = P4; Grey diamond = M1; Grey rectangle = M2; Grey triangle = M3. Line represents y = x. Pearson's correlations: (**A**) *T. terrestris*: n = 76, *R* = 0.77, *P*<0.001; *T. bairdii*: n = 32, *R* = 0.76, *P*<0.001; *T. indicus*: n = 12, *R* = 0.86, *P*<0.001; (**B**) *T. terrestris*: n = 74, *R* = 0.86, *P*<0.001; *T. bairdii*: n = 29, *R* = 0.89, *P*<0.001; *T. indicus*: n = 12, *R* = 0.90, *P*<0.001; (**C**) *T. terrestris*: n = 70, *R* = 0.86, *P*<0.001; *T. bairdii*: n = 24, *R* = 0.90, *P*<0.001; *T. indicus*: n = 12, *R* = 0.91, *P*<0.001. The line denotes y = x.

Comparing the RLD of $P^3$–$M^2$ on the basis of the RLD of the $M^3$ (as the newest tooth) of the respective animals from natural habitats or zoos (Fig 21), *T. terrestris* again showed significant differences, with zoo animals having higher RLD for the respective $M^3$ RLD. By contrast, no differences between animals from natural habitats and zoos were evident for any tooth in *T. indicus*.

## Discussion

This study suggests that the masticatory movement of tapirs is, in comparison to other perissodactyls, mainly orthal and not lateral. This movement, paired with anisodonty, leads to a characteristic abrasion pattern mainly on the lingual part of the upper cheek teeth, which allows quantifying the progress of wear. The lophodont tooth structure, with enamel-covered and basin-filled lophs, leads to a consistent increase of the dentine area with progressing wear, again allowing quantifying wear. Wear stages correspond to the eruption sequence of cheek teeth, and mesowear scoring appears feasible on the buccal aspect of tapir cheek teeth. Generally, individuals kept in zoos showed slightly less, not more, wear than free-ranging conspecifics.

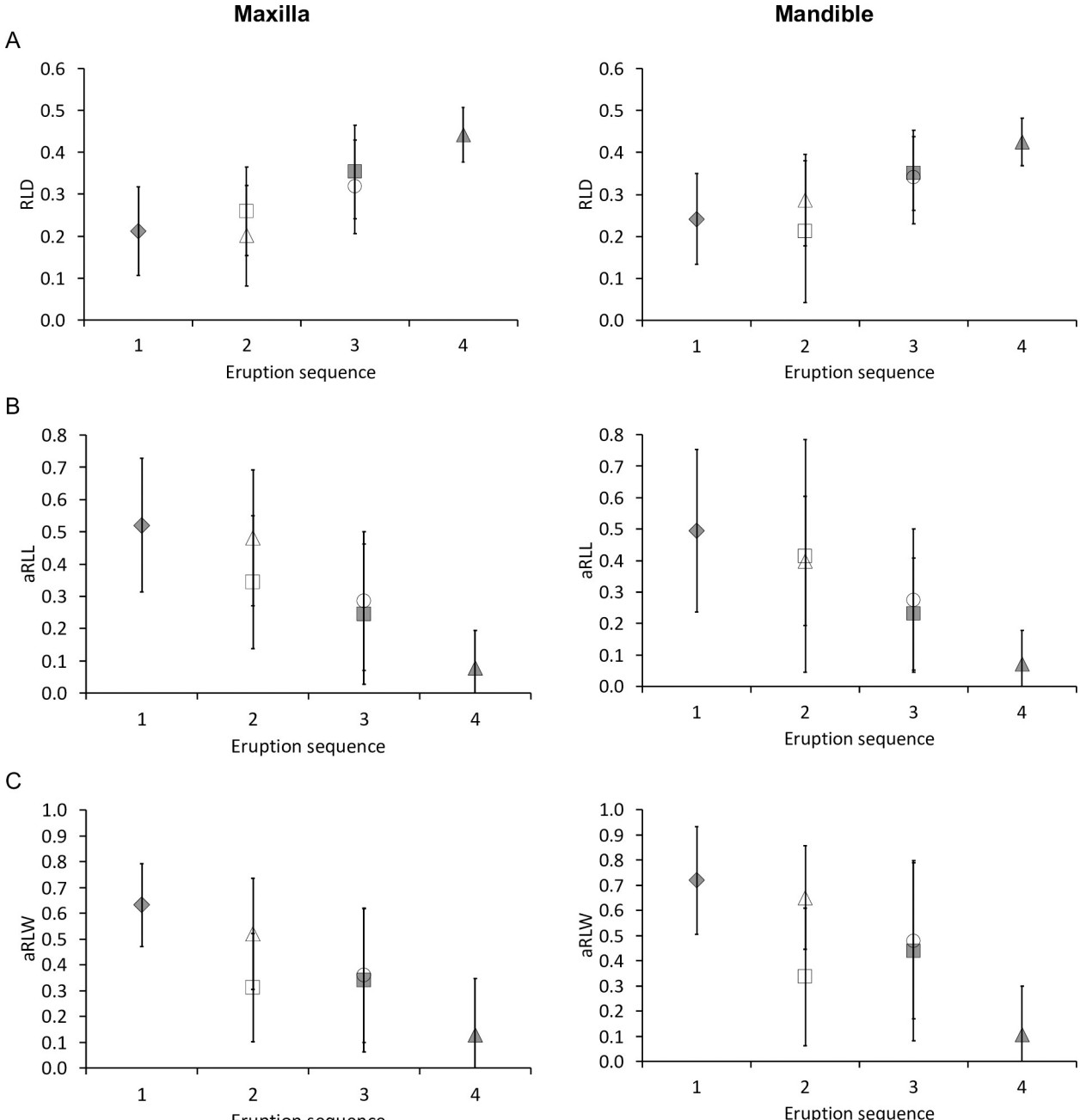

**Fig 11. Tapir tooth wear and eruption sequence.** Relationship of the eruption sequence of maxillary (left) and mandibular (right) cheek teeth of three tapir species with mean (± SD) quantitative wear measures of the total sample: **(A)** relative loph distance (RLD), **(B)** added relative loph length (aRLL), **(C)** added relative loph width (aRLW), indicating that generally, teeth that erupted earlier had more wear. White rectangle = P2; White triangle = P3; White circle = P4; Grey diamond = M1; Grey rectangle = M2; Grey triangle = M3. Pearson's correlations: **(A)** *T. terrestris*: maxilla n = 268, R = 0.58, P<0.001; mandible n = 77, R = 0.49, P<0.001; *T. bairdii*: maxilla n = 35, R = 0.42, P = 0.011; mandible n = 33, R = 0.43, P = 0.013; *T. indicus*: maxilla n = 88, R = 0.55, P<0.001; mandible n = 12, R = 0.62, P = 0.030; **(B)** *T. terrestris*: maxilla n = 267, R = -0.56, P<0.001; mandible n = 77, R = -0.47, P<0.001; *T. bairdii*: maxilla n = 34, R = -0.53, P = 0.001; mandible n = 33, R = -0.49, P = 0.004; *T. indicus*: maxilla n = 86, R = -0.61, P<0.001; mandible n = 12, R = -0.59, P = 0.042; **(C)** *T. terrestris*: maxilla n = 271, R = -0.55, P<0.001; mandible n = 76, R = -0.59, P<0.001; *T. bairdii*: maxilla n = 34, R = -0.35, P = 0.042; mandible n = 31, R = -0.47, P = 0.007; *T. indicus*: maxilla n = 89, R = -0.57, P<0.001; mandible n = 12, R = -0.62, P = 0.031.

**Maxilla** **Mandible**

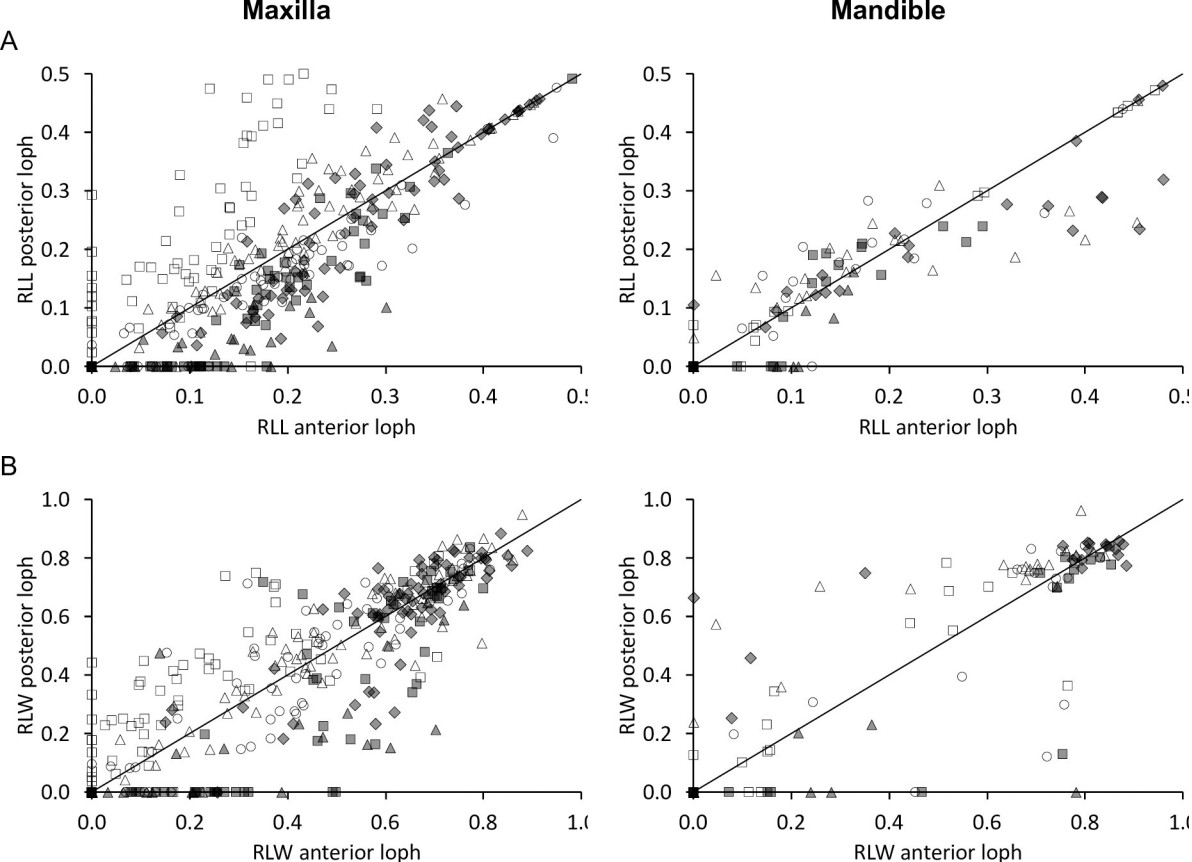

**Fig 12. Tapir tooth wear–anterior vs posterior lophs. (A)** Relative loph length (RLL) and **(B)** relative loph width (RLW) of cheek teeth of three tapir species, comparing the anterior and posterior loph of individual teeth. White rectangle = P2; White triangle = P3; White circle = P4; Grey diamond = M1; Grey rectangle = M2; Grey triangle = M3. Symbols on the y = x line represent teeth where dentine basins of the lophs have confluated and values were calculated by dividing the joint loph length by 2. Pearson's correlations: **(A)** *T. terrestris*: maxilla n = 267, *R* = 0.79, *P*<0.001; mandible n = 77, *R* = 0.91, *P*<0.001; *T. bairdii*: maxilla n = 34, *R* = 0.90, *P*<0.001; mandible n = 32, *R* = 0.92, *P*<0.001; *T. indicus*: maxilla n = 86, *R* = 0.80, *P*<0.001; mandible n = 12, *R* = 0.98, *P*<0.001; **(B)** *T. terrestris*: maxilla n = 271, *R* = 0.89, *P*<0.001; mandible n = 73, *R* = 0.81, *P*<0.001; *T. bairdii*: maxilla n = 34, *R* = 0.88, *P*<0.001; mandible n = 26, *R* = 0.93, *P*<0.001; *T. indicus*: maxilla n = 87, *R* = 0.83, *P*<0.001; mandible n = 12, *R* = 0.83, *P*<0.001. The line denotes y = x.

The main limitation of the present study was the small number of specimens available for *T. bairdii* (n = 6) and free-ranging *T. indicus* (n = 6). On the other hand, 29 *T. terrestris* specimens from natural habitats and 18 from zoos were available. To facilitate comparisons across species (that show some size differences), we introduced relative values, which are not dependent of the absolute size of teeth. This allowed us to pool all individuals together when comparing relative values.

## Mastication

Transferring from the interplay of dental anatomy and chewing direction from other species such as elephants, rodents or macropods [4], the mediolateral orientation of the lophs of tapir cheek teeth (Fig 1) would lead to the expectation of propalinal rather than lateral chewing movements. Our analysis of chewing in lowland tapirs led to the conclusion that the masticatory movement in this species is orthal without any lateral movement, or alternatively with only such a slight lateral component that it is readily missed during observation. A propalinal movement was not observed. Ryder [8] observed tapirs while feeding on hay and fresh grass,

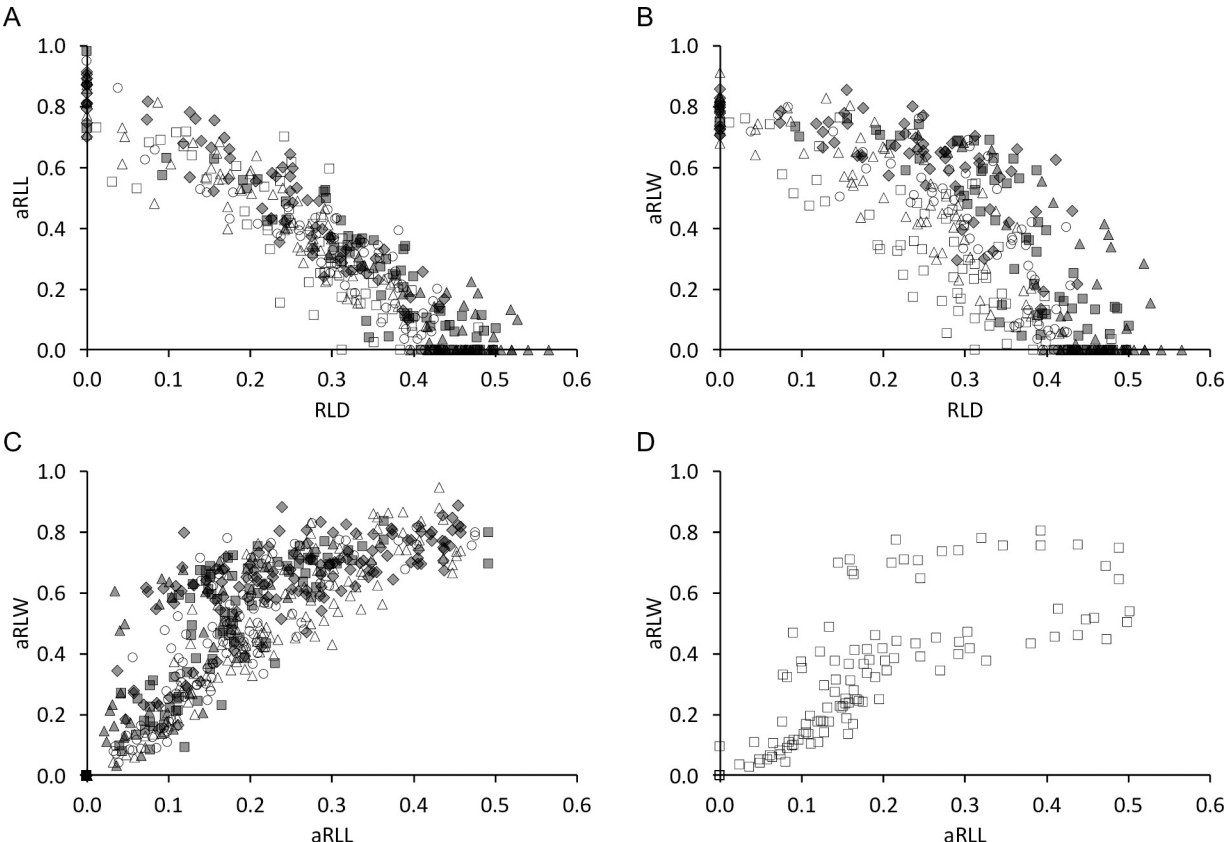

**Fig 13. Tapir tooth wear–correlations of quantitative measures maxilla.** Correlations between quantitative wear measures in maxillary cheek teeth of three tapir species **(A)** relative loph distance (RLD) and added relative loph length (aRLL); **(B)** RLD and added relative loph width (aRLW); **(C)** and **(D)** aRLL and aRLW. White rectangle = P2; White triangle = P3; White circle = P4; Grey diamond = M1; Grey rectangle = M2; Grey triangle = M3. Pearson's correlations: **(A)** *T. terrestris*: n = 266, *R* = -0.89, *P*<0.001; *T. bairdii*: n = 34, *R* = -0.93, *P*<0.001; *T. indicus*: n = 86, *R* = -0.91, *P*<0.001; **(B)** *T. terrestris*: n = 266, *R* = -0.81, *P*<0.001; *T. bairdii*: n = 34, *R* = -0.86, *P*<0.001; *T. indicus*: n = 85, *R* = -0.83, *P*<0.001; **(C and D)** *T. terrestris*: n = 267, *R* = 0.82, *P*<0.001; *T. bairdii*: n = 34, *R* = 0.87, *P*<0.001; *T. indicus*: n = 85, *R* = 0.83, *P*<0.001.

came to a similar conclusion with very little lateral movement, and even stated that the biomechanical properties of ingesta do not have an impact on tapir jaw movement. Chang [41] suggested that lophodont teeth, as found in tapirs, are rather useful for crushing than grinding, and thus assumed lophodont species to chew in an orthal motion. The analysis of masticatory movement by visual observations is clearly limited [42]. We share this opinion based on two own observations. First, when observing tapirs from a lateral view, the bulging of the visible cheek during the power stroke can easily lead to the impression of a lateral power stroke component. By contrast, in a frontal view, this bulging was clearly symmetrical on both cheeks, and no mandibular excursion was observed. Secondly, even in frontal view, movement of the prominent proboscis of tapirs can also impair judgement based on the visual appearance. The comparison with related species (domestic horse and greater one-horned rhinoceros), where the masticatory motion is well known, showed obvious differences (Fig 3). Investigation of striation on the facets of cheek teeth is considered more exact for the reconstruction masticatory movements, compared to visual observation [42]. Such an analysis has not yet been performed on extant tapir species, but von Koenigswald [9] suggested that the mastication pattern of *Tapirus* is similar to the one of the fossil *Lophoidon*, in which he determined some movement in a mesiolingual direction. Notably, these measurements were performed on isolated tooth rows, not on complete skulls.

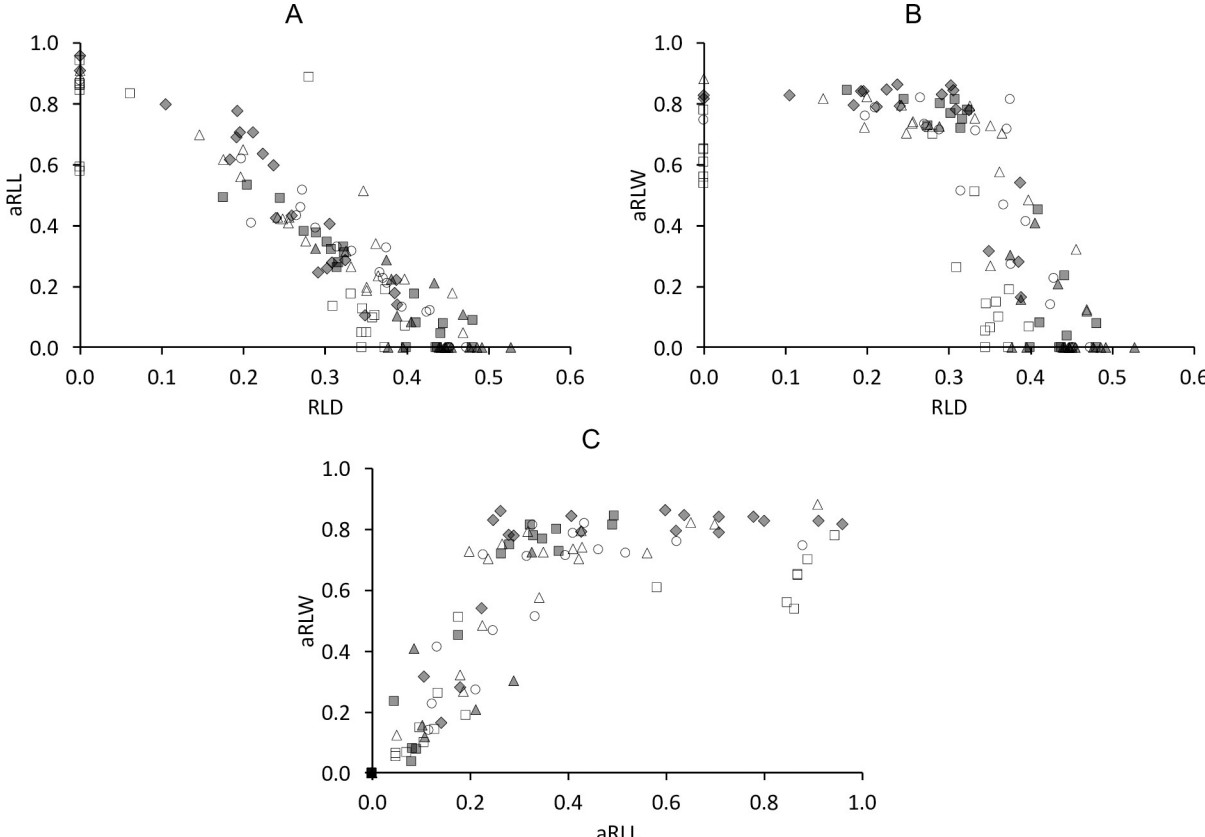

**Fig 14. Tapir tooth wear–correlations of quantitative measures mandible.** Correlations between quantitative wear measures in mandibular cheek teeth of three tapir species **(A)** relative loph distance (RLD) and added relative loph length (aRLL); **(B)** RLD and added relative loph width (aRLW); **(C)** aRLL and aRLW. White rectangle = P2; White triangle = P3; White circle = P4; Grey diamond = M1; Grey rectangle = M2; Grey triangle = M3. Pearson's correlations: **(A)** *T. terrestris*: n = 77, *R* = -0.90, *P*<0.001; *T. bairdii*: n = 32, *R* = -0.93, *P*<0.001; *T. indicus*: n = 12, *R* = -0.92, *P*<0.001; **(B)** *T. terrestris*: n = 76, *R* = -0.67, *P*<0.001; *T. bairdii*: n = 30, *R* = -0.64, *P*<0.001; *T. indicus*: n = 12, *R* = -0.63, *P* = 0.027; **(C)** *T. terrestris*: n = 76, *R* = 0.79, *P*<0.001; *T. bairdii*: n = 31, *R* = 0.75, *P*<0.001; *T. indicus*: n = 12, *R* = 0.82, *P* = 0.001.

Our second line of circumstantial evidence against distinct lateral or propalinal chewing movements comes from the–admittedly subjective–manipulation of a tapir skull, in which we did not achieve lateral mandibular excursion without partial loss of occlusal contact along the cheek tooth row. To our experience, in species with a clear lateral chewing motion such as horses or cattle, the mandible can be manipulated for a lateral excursion with the whole cheek tooth row in occlusal contact. Another feature of the tapir dentition, the prominent interlocking of the I$^3$ and the I$_3$ and lower canine during occlusion (Fig 3), also speaks against a prominent lateral motion. Manipulation of the skull imitating propalinal chewing was not possible.

Our third line of evidence derives from another striking detail of the tapir skull, especially in *T. terrestris*—the distinctive sagittal crest. Holbrook [43] stated that the tapirs' temporal muscle, which has its insertion on the facet of this prominent sagittal crest, was large and filled the whole temporal fossa. In comparison, the temporal muscle is prominent in carnivorous species, where no lateral chewing is present, and is responsible for a snapping movement [44, 45]. A sagittal crest is prominent in rodents that require a strong incisor bites, such as in chisel-tooth digging burrowers or species that are particularly lignivorous such as beavers [46]. In herbivorous species with a lateral chewing motion, the *M. masseter* is the functionally more prominent chewing muscle [44, 45].

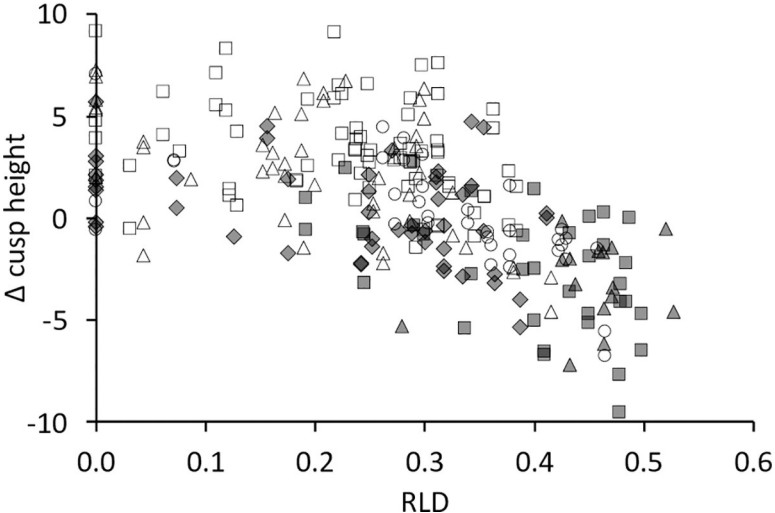

**Fig 15. Tapir tooth wear–cusp height changes.** Relationship between the Δ cusp height (the difference of buccal–lingual cusp height) and wear (quantified as the relative loph distance RLD) in three tapir species. White rectangle = P2; White triangle = P3; White circle = P4; Grey diamond = M1; Grey rectangle = M2; Grey triangle = M3. Pearson's correlations: *T. terrestris*: n = 108, *R* = -0.61, *P*<0.001; *T. indicus*: n = 35, *R* = -0.58, *P*<0.001; Spearman's correlation: *T. bairdii*: n = 4, *R* = 0.20, *P* = 0.917.

Our fourth line of evidence for orthal chewing in tapirs follows the logic of Fortelius [31], who stated that, judging from tapir tooth morphology, tapirs have relatively little lateral chewing motion. A high degree of anisodonty between the maxillary and the mandibular teeth, as observed in tapirs as well as other species, e.g. horses, is expected to lead to an even wear of the whole occlusal surface of the larger tooth if a full lateral chewing motion ensures similar attrition on the buccal and the lingual cusps. This is evidently the case in horses. During an orthal motion, those cusps of the larger tooth that are not opposed by the smaller tooth will experience less wear, and hence become more prominent over time. The consistent change in Δ cusp height with progressing wear (Fig 15) supports this scenario in tapirs: As the maxillary and mandibular teeth are perfectly aligned to each other on the lingual side (Fig 4), the lingual parts of the maxillary teeth will be worn down faster than the buccal parts.

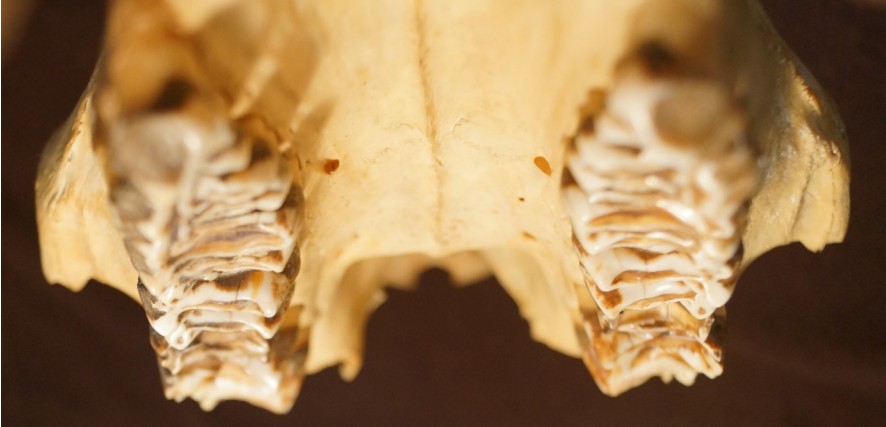

**Fig 16. Tapir maxillary teeth–cusp height gradient.** Cranial view onto the maxillary tooth rows of a *Tapirus indicus*, showing the typical elevated shape of the buccal side ('ectoloph') that appears to develop during wear in most specimens investigated.

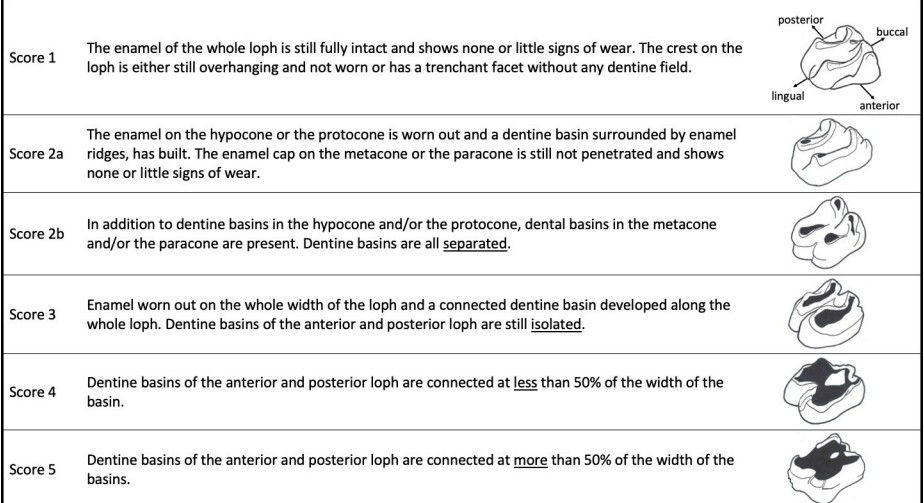

| | |
|---|---|
| Score 1 | The enamel of the whole loph is still fully intact and shows none or little signs of wear. The crest on the loph is either still overhanging and not worn or has a trenchant facet without any dentine field. |
| Score 2a | The enamel on the hypocone or the protocone is worn out and a dentine basin surrounded by enamel ridges, has built. The enamel cap on the metacone or the paracone is still not penetrated and shows none or little signs of wear. |
| Score 2b | In addition to dentine basins in the hypocone and/or the protocone, dental basins in the metacone and/or the paracone are present. Dentine basins are all separated. |
| Score 3 | Enamel worn out on the whole width of the loph and a connected dentine basin developed along the whole loph. Dentine basins of the anterior and posterior loph are still isolated. |
| Score 4 | Dentine basins of the anterior and posterior loph are connected at less than 50% of the width of the basin. |
| Score 5 | Dentine basins of the anterior and posterior loph are connected at more than 50% of the width of the basins. |

**Fig 17. Tapir tooth wear score.**

## Dental anatomy

Tapir premolars and molars are molariform and highly bilophodont [7]. The (anterior) metaloph connects the (lingual) protocone and (buccal) paracone, while the (posterior) protoloph connects the (lingual) hypocone and (buccal) metacone. The ectoloph (connecting the paracone and the metacone), a prominent feature in rhinoceroses, is of small importance in tapir teeth, remaining mostly on the $P^1$ and $P_2$ [7]. $P_1$ is absent, but $P_2$ is, compared to the other lower cheek teeth, elongated (Fig 5), while the remaining lower cheek teeth are more or less consistent in their length. Regarding the upper tooth row, $M^2$ and $M^3$ seem to be the longest teeth, while $P^1$ and $P^2$ are generally the shortest cheek teeth. The $P^1$ has a triangular shape and is the only cheek tooth that is not bilophodont; therefore, only one loph was measurable and presented.

Fortelius [31] investigated upper and lower cheek teeth of different species regarding anisodonty and introduced the anisodonty index (ADI), which we calculated with our samples as well. Tapirs are generally anisodont on all cheek teeth, with the upper cheek teeth consistently wider than the lower ones. The ADI is highest for the most anterior cheek teeth, and becomes systematically smaller towards the back of the tooth row. The ADI between different tapir species is comparable, another reason why we assume dental morphology to be similar between the examined species. The denoted ADI (M2) of Fortelius [31] is slightly higher than our calculated ADI (1.44 vs. 1.26–1.28). This might be due to the fact that he measured the tooth width as the maximum basal width (that differs more between maxillary and mandibular teeth), while we used the average loph width. As explained further above, with progressed wear, this anisodonty leads to a buccal overhang, which is detectable and very characteristic in all examined species.

Comparing the width of the anterior and posterior loph, the molars are broadening towards their anterior part. By contrast, the premolar teeth are rather broadening towards the posterior loph or have a constant width, except $P^4$, which more resembles the shape of the molar teeth. The lower cheek teeth show similar features; the molar teeth are broader on the anterior loph and the premolars are broader on the posterior loph, including $P_4$. Comparing the anterior loph width with tooth length, there is a correlation along all teeth, meaning the anterior loph has a constant width in every cheek tooth compared to its length (Fig 8). In comparison, the

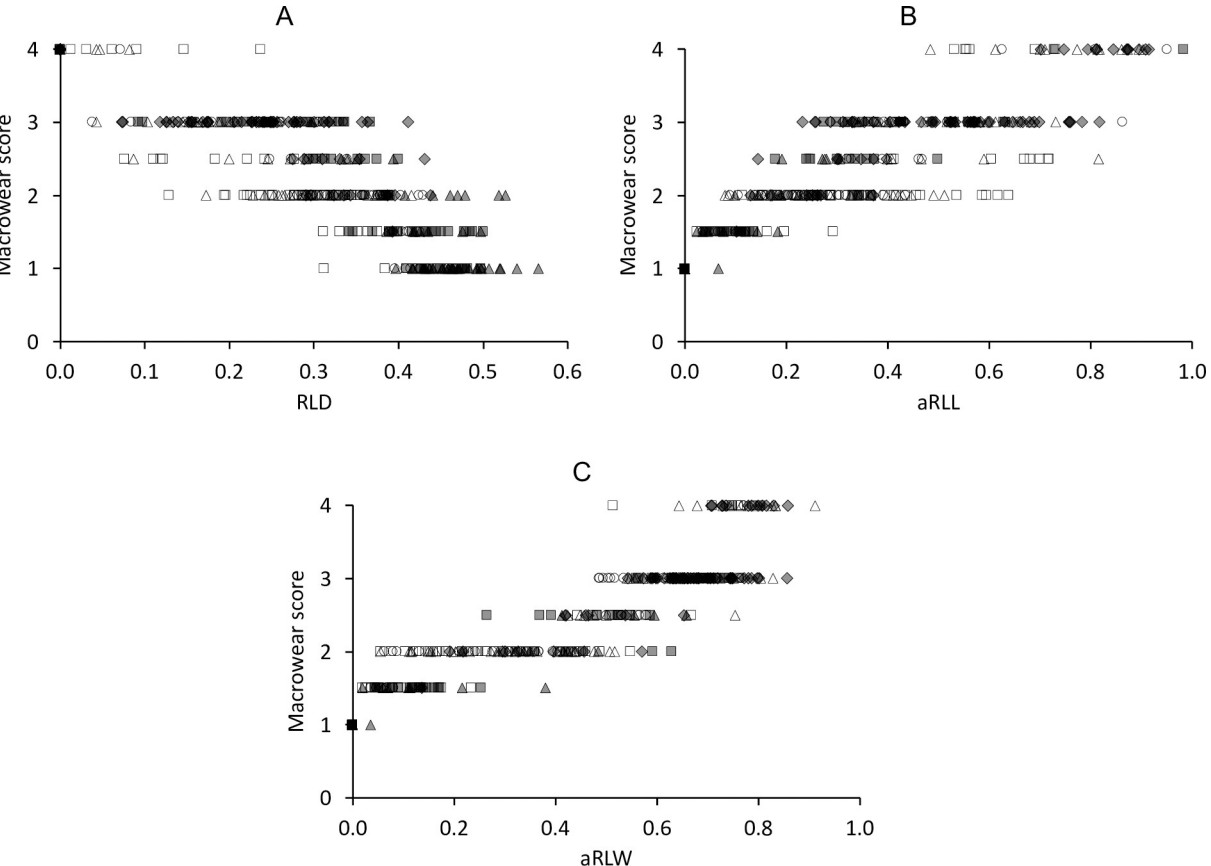

**Fig 18. Tapir tooth wear–quantitative measures and macrowear score.** Correlations between quantitative wear measures and subjective macrowear score of maxillary cheek teeth in three tapir species **(A)** relative loph distance (RLD) and Macrowear score; **(B)** added relative loph length (aRLL) and Macrowear score; **(C)** added relative loph width (aRLW) and Macrowear score; White rectangle = P2; White triangle = P3; White circle = P4; Grey diamond = M1; Grey rectangle = M2; Grey triangle = M3. Spearman's correlations: **(A)** *T. terrestris*: n = 268, *R* = -0.84, *P*<0.001; *T. bairdii*: n = 35, *R* = -0.92, *P*<0.001; *T. indicus*: n = 88, *R* = -0.86, *P*<0.001; **(B)** *T. terrestris*: n = 267, *R* = 0.81, *P*<0.001; *T. bairdii*: n = 34, *R* = 0.92, *P*<0.001; *T. indicus*: n = 86, *R* = 0.76, *P*<0.001; **(C)** *T. terrestris*: n = 271, *R* = 0.93, *P*<0.001; *T. bairdii*: n = 34, *R* = 0.93, *P*<0.001; *T. indicus*: n = 89, *R* = 0.92, *P*<0.001.

posterior loph width does not change proportionately with tooth length, so that the posterior loph of upper premolars is proportionally wider than the one of upper molars (Fig 9). This might be due to the fact that the molar teeth erupt consecutively while the individual is still growing, and that at the point of eruption, less space is available in the tooth socket on the posterior end compared to the anterior end (Fig 22). And as the M1 is the first permanent cheek tooth [30], the permanent premolars, which erupt later, have more space on the posterior end compared to the molar teeth.

Simpson [47] performed similar analyses, comparing the maximum loph width with tooth length and received results comparable to our own (Fig 23). The data indicate a difference in size, suggesting that some individuals studied by Simpson [47] were larger than those available to us.

## Tapir tooth wear

We used the presence and dimension of dentine basins of each loph as reference to classify tooth wear for each loph. A freshly erupted tooth comprises two horizontally aligned lophs, which are, as in almost any mammal tooth at eruption, fully covered by the enamel cap. They

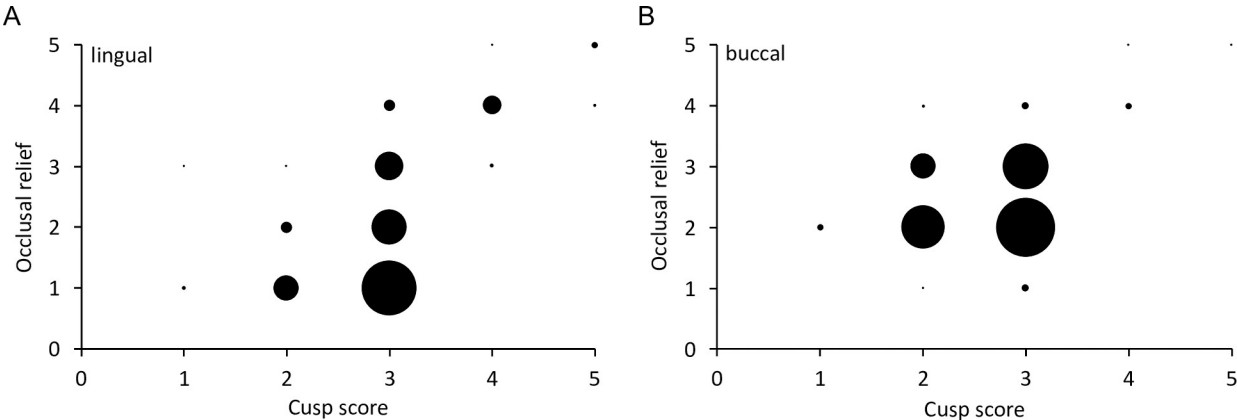

**Fig 19. Tapir mesowear scores–cusp shape and occlusal relief.** Relationship of the two mesowear scores, the cusp shape and the occlusal relief for (**A**) lingual and (**B**) buccal aspects of maxillary cheek teeth of three tapir species. The size of the dots is proportional to the number of cases represented. Spearman's correlations: (**A**) *T. terrestris*: n = 270, $R = 0.57$, $P<0.001$; *T. bairdii*: n = 35, $R = 0.16$, $P = 0.351$; *T. indicus*: n = 90, $R = 0.65$, $P<0.001$; (**B**) *T. terrestris*: n = 272, $R = 0.22$, $P<0.001$; *T. bairdii*: n = 35, $R = 0.19$, $P = 0.282$; *T. indicus*: n = 90, $R = 0.36$, $P<0.001$.

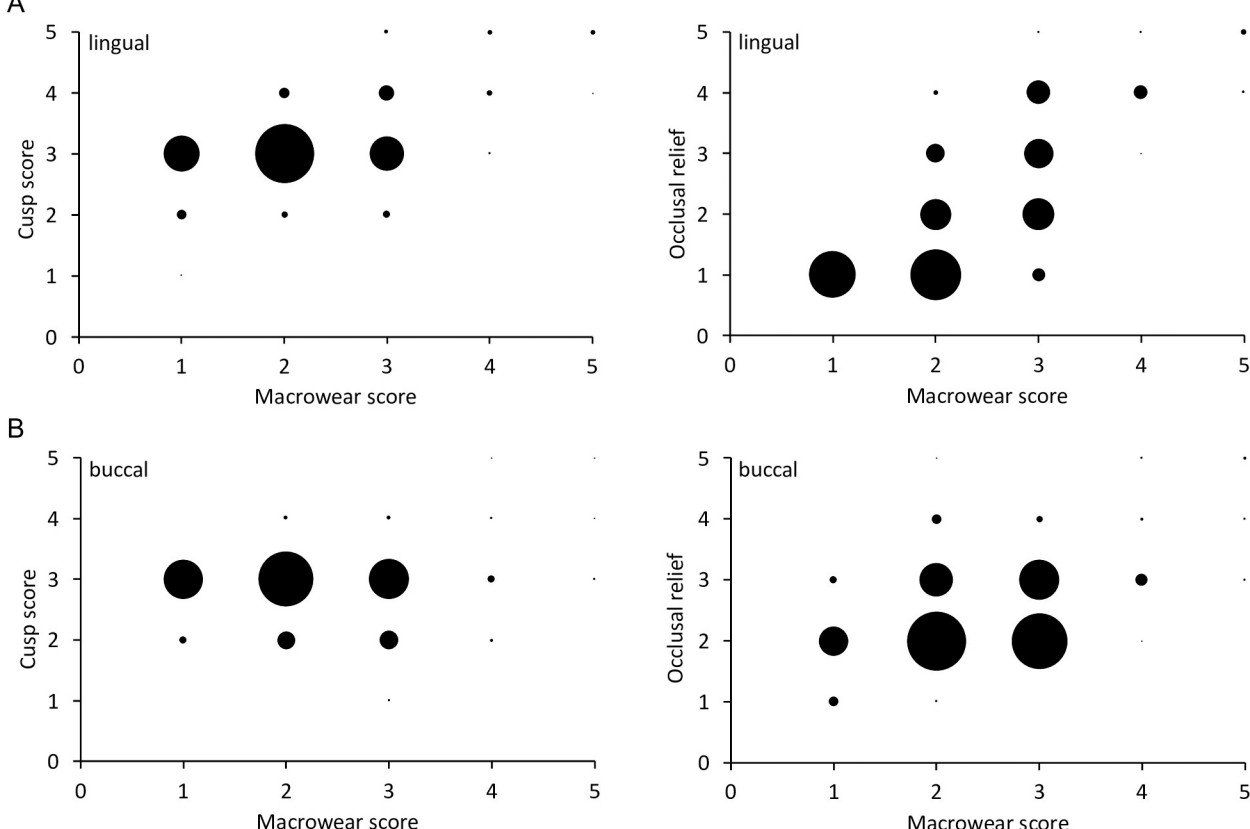

**Fig 20. Tapir mesowear score–correlation with macrowear scores.** Relationship of two mesowear scores (cusp shape, occlusal relief) and the macrowear score (as a measure of 'wear stage') for (**A**) lingual and (**B**) buccal aspects of maxillary cheek teeth of three tapir species. The size of the dots is proportional to the number of cases represented. Spearman's correlations: (**A**) *T. terrestris*: cusp shape n = 276, $R = 0.44$, $P<0.001$; occlusal relief n = 270, $R = 0.77$, $P<0.001$; *T. bairdii*: cusp shape n = 35, $R = 0.08$, $P = 0.665$; occlusal relief n = 35, $R = 0.80$, $P<0.001$; *T. indicus*: cusp shape n = 92, $R = 0.29$, $P = 0.005$; occlusal relief n = 91, $R = 0.60$, $P<0.001$; (**B**) *T. terrestris*: cusp shape n = 276, $R = -0.08$, $P = 0.202$; occlusal relief n = 272, $R = 0.41$, $P<0.001$; *T. bairdii*: cusp shape n = 35, $R = -0.00$, $P = 0.996$; occlusal relief n = 35, $R = 0.18$, $P = 0.291$; *T. indicus*: cusp shape n = 92, $R = -0.00$, $P = 0.977$; occlusal relief n = 90, $R = 0.44$, $P<0.001$.

**Table 3. Tapir mesowear scores.**

| Species | | Habitat | P³ | P⁴ | M¹ | M² | M³ |
|---|---|---|---|---|---|---|---|
| *T. terrestris* | cusp score | Natural | 2.95 (2;5) [n] = 22 | 2.63 (2;4) [n] = 27ᵃ | 2.55 (2;4) [n] = 22ᴬ | 2.52 (1;3) [n] = 29 | 2.81 (2;3) [n] = 26 |
| | | Zoo | 2.76 (2;3) [n] = 17 | 2.33 (1;3) [n] = 18ᵇ | 2.00 (1;3) [n] = 17ᴮ | 2.39 (1;3) [n] = 18 | 2.81 (2;3) [n] = 17 |
| | occlusal relief | Natural | 2.68 (2;5) [n] = 22ᵃ | 2.33 (1;4) [n] = 27ᵃ | 2.55 (2;4) [n] = 22ᴬ | 2.28 (1;3) [n] = 29 | 2.00 (1;3) [n] = 24 |
| | | Zoo | 2.41 (2;3) [n] = 17ᵇ | 2.06 (1;3) [n] = 18ᵇ | 2.18 (2;3) [n] = 17ᴮ | 2.00 (1;3) [n] = 18 | 1.81 (1;3) [n] = 16 |
| *T. indicus* | cusp score | Natural | 2.75 (2;3) [n] = 4 | 2.80 (2;3) [n] = 5 | 2.50 (2;3) [n] = 2 | 1.80 (1;3) [n] = 5 | 2.67 (2;4) [n] = 6 |
| | | Zoo | 2.88 (1;4) [n] = 8 | 2.70 (2;3) [n] = 10 | 2.63 (1;3) [n] = 8 | 2.22 (1;3) [n] = 9 | 2.60 (1;3) [n] = 10 |
| | occlusal relief | Natural | 2.25 (2;3) [n] = 4 | 2.00 (2;2) [n] = 5 | 2.50 (2;3) [n] = 2 | 2.00 (2;2) [n] = 5 | 2.33 (2;4) [n] = 6 |
| | | Zoo | 2.25 (2;3) [n] = 8 | 2.80 (2;4) [n] = 10 | 2.25 (2;3) [n] = 8 | 2.22 (2;3) [n] = 9 | 1.89 (1;2) [n] = 9 |

Average mesowear score (min value; max value) of the buccal cusp and occlusal relief, regarding only specimens with macrowear score between one and three. Cusp score: 1 = sharp; 2 = round-sharp; 3 = round; 4 = round-round; 5 = blunt. Occlusal relief: 1 = high-high; 2 = high; 3 = high-low; 4 = low; 5 = flat-negative.

Different superscripts within columns and mesowear scores indicate significant differences ᴬᴮ(P<0.050) or trends ᵃᵇ(P<0.081) between animals from natural habitats and zoos.

are aligned in a buccolingual direction and their individual cross section (in anterioposterior direction) has a triangular shape along the whole loph, cuspidal to its crest. With ongoing wear, the lophs 'collapse' (visualized in Fig 5 of [9]), the enamel is worn off and the underlying dentine appears in terms of a dentine basin surrounded by enamel ridges. This happens first on the lingual part of the tooth, most likely due to the anisodonty and the orthal chewing motion, as explained above. The dentine basins expand in length and width, until the basins of the two lophs of a tooth meet, which also most likely occurs first on the lingual part of the tooth. This connection becomes broader with progressing wear, until one uniform dentine basin is present covering the occlusal surface. This progress is reflected in the macrowear scoring system (Fig 17). To validate this categorical score, we used several wear measurements (RLD, aRLL, aRLW), which correlated with the macrowear score (Fig 18). While a high macrowear score represents more wear, the RLD becomes smaller with increasing wear, as it describes the minimum distance between two dentine basins (or loph crests if no dentine basin is present) in relation to its tooth length. Single outliers are present at score 4. In these individual teeth, the connection of dentine basins occurred on the buccal part of the tooth, while the loph distance, resulting in the RLD, was constantly measured on the lingual part, where, in these few cases, the dentine basins were separated. When comparing aRLL and aRLW, the values correlate for P³ to M³, meaning the dimensions of the dentine basins progress in a predictable manner (Fig 13). There is less correlation for P² (Fig 13), meaning the dimension of the dentine basins can strongly differ between individuals, which complicates the classification of wear for this tooth.

The dimensions of dentine basins of the lower tooth row progress in slightly differently (Fig 14). The dentine basins first expand mainly in a buccolingual direction and secondary in an anterioposterior direction. This is explained by the fact that the narrower mandibular cheek teeth occlude along their whole width. Once the maximum width of the dentine basin has

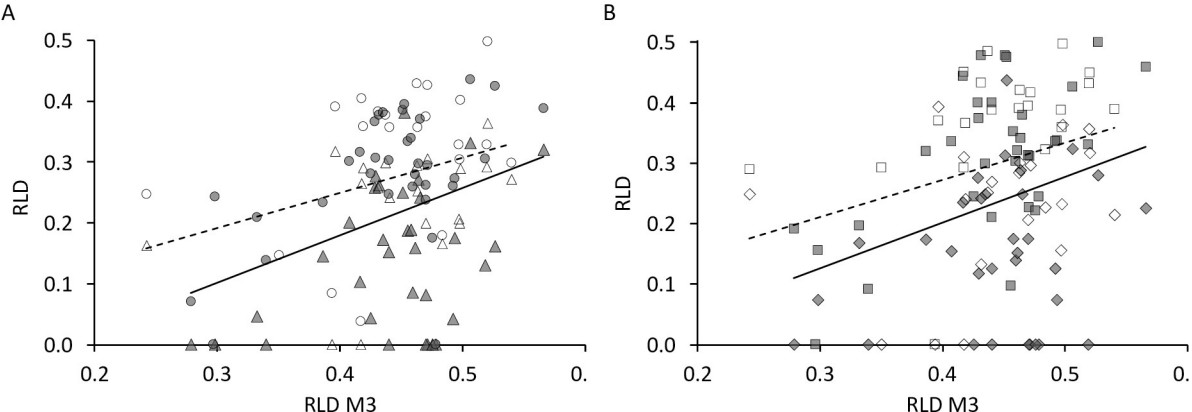

**Fig 21. Tapir tooth wear–natural habitat vs zoo.** Comparison of the relative loph distance (RLD, an objective measure of wear) of selected teeth and RLD of M$^3$ distinguished between free-ranging and zoo individuals; Grey = free-ranging; White = zoo; Triangles = P$^3$; Diamonds = M$^1$; Black line = free-ranging; Dashed line = captive. **(A)** Triangles = P$^3$; Circles = P$^4$; **(B)** Diamonds = M$^1$; Rectangles = M$^2$;. Differences between free-ranging and captive individuals were significant (assessed by general linear models) for *T. terrestris*: P$^3$ (n = 36, $P < 0.001$), P$^4$ (n = 37, $P = 0.026$), M$^1$ (n = 37, $P = 0.001$), M$^2$ (n = 37, $P = 0.021$); *T. indicus*: no significant differences—P$^3$ (n = 12, $P = 0.749$), P$^4$ (n = 13, $P = 0.584$), M$^1$ (n = 11, $P = 0.495$), M$^2$ (n = 13, $P = 0.564$).

developed, the expansion continues in an anterioposterior direction until the whole occlusal surface is occupied by the dentine basin. Therefore, the RLW of the upper and lower cheek teeth correlate weakly (Fig 10). Although there seems to be quite some variability in the relative sequence of RLW between maxillary and mandibular teeth, the majority of mandibular teeth reaches its maximum RLW before the maxillary ones (Fig 10). The aRLW of P$_2$ does not reach as high values as the other mandibular cheek teeth, as it is the only lower cheek tooth with a slightly developed ectoloph [7], which prevents the dentine basin from expanding across the whole width of the tooth. But the wear pattern of P$_2$ is still rather definable compared to P$^2$. The values RLD, aRLW, aRLD also correlate in the mandibular tooth row, but the relationship shows a different pattern, due to the reduced width of the mandibular teeth. Comparing RLD, RLL and RLW between the maxillary and mandibular tooth row, simple associations are evident for the measurements relating to the teeth's length (RLD, RLL) but not for the one relating to their width (RLW) (Fig 10). Considering the teeth's longitudinal axis, wear progresses

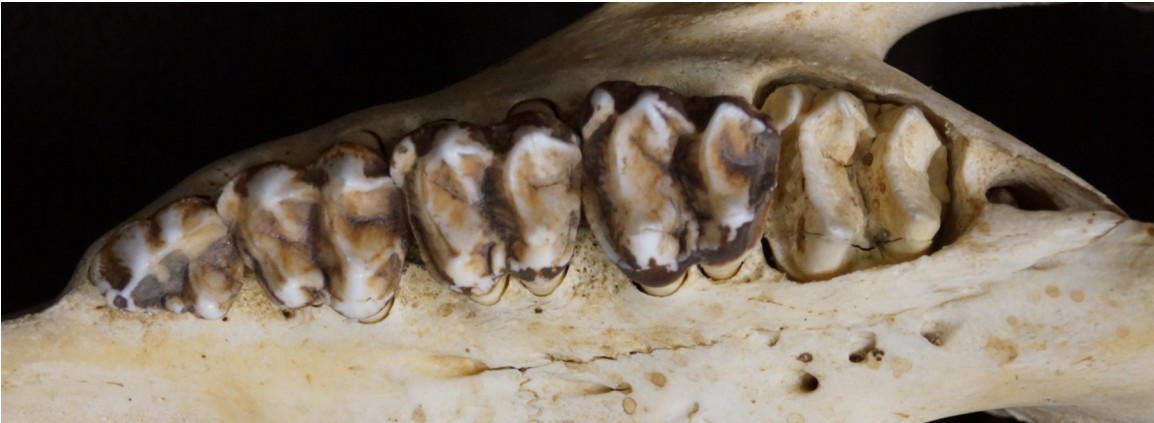

**Fig 22. Tapir maxilla–immature stage.** Example of a tapir in which the M$^2$ (in the photo, the tooth farthest to the right) is in the process of eruption. Note that the space available for the tooth becomes smaller towards the posterior (here, right) part of the jaw. The shape of the M$^1$ (second from right) probably indicates the conditions of a shorter jaw at the time of its own eruption.

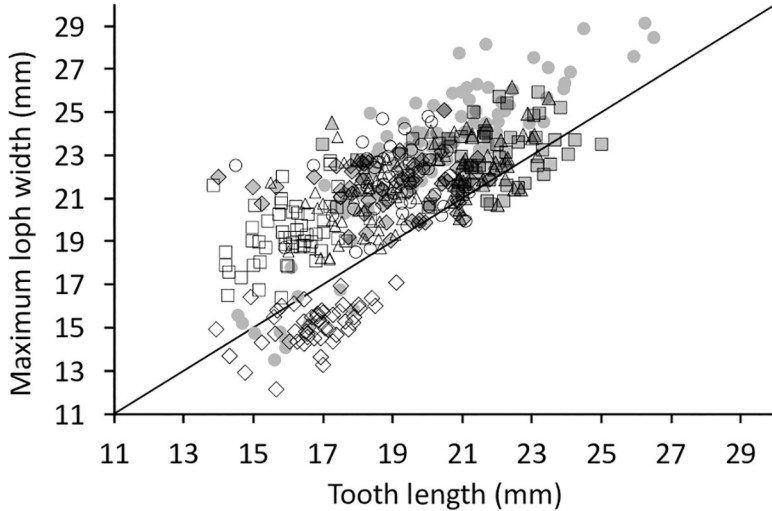

**Fig 23. Tapir teeth–comparison with literature data.** Tooth width compared to the maximum loph width of the upper jaw in *T. terrestris*. White rectangle = P2; White triangle = P3; White circle = P4; Grey diamond = M1; Grey rectangle = M2; Grey triangle = M3; Grey circles = Data from Simpson [47; read from graph]. The line denotes y = x.

uniformly between the upper and lower tooth row. Given that wear is an accumulative process, it is self-evident that measures of wear should, in the population average, be more prominent in teeth that erupt earlier (Fig 11).

Maffei [32] and Gibson [33] developed keys to determine tapirs ages by tooth wear based on hunted free-ranging individuals. As we mostly had no information about age from our specimens, we cannot compare our material to theirs. The only skull with identified age in our study was labelled as 10 years old and had non-occluding third molars, which is contradiction to the age keys of Maffei [32] and Gibson [33] (wear on M3 present at 8 years). We hope that the more detailed anatomical description of macrowear stages provided in Fig 17, and the possibility of quantifying wear using objective measurements (RLD, RLL, RLW), will facilitate more detailed future studies on relationships of tapir tooth wear and age. In other herbivores, tooth wear is often quantified as a loss of dental substance in mm over time [48]. Even if individual ages of animals were known, this would be hardly applicable to tapirs, as the loss of tooth substance is uneven along the width of maxillary cheek teeth.

For macropods, which also have bilophodont cheek teeth [4, 49], McArthur and Sanson [50] developed a comparable macrowear score and used it to perform comparisons between different populations of eastern grey kangaroos (*Macropus giganteus*) and western grey kangaroos (*Macropus fuliginosus*). Thus, using dental basin dimensions for subjective scoring or objective quantification appears a suitable method to measure tooth wear in biolophodont mammals. Note that in macropods, however, dental basins develop more evenly along the lophs [50], rather than mainly on the lingual side, corresponding to the macropods' lower anisodonty index (1.18) compared to tapirs (1.44) [31]. Additionally, macropods may have a more pronounced lateral component in their chewing pattern [49]. Therefore, they do not appear to show the change in the cusp height between lingual and buccal cusps characteristic of tapirs.

### Tapir mesowear

The mesowear method is mainly used to infer the natural diet from occlusal cross morphology of worn teeth. It was introduced by Fortelius and Solounias [2] and later revised by Kaiser and

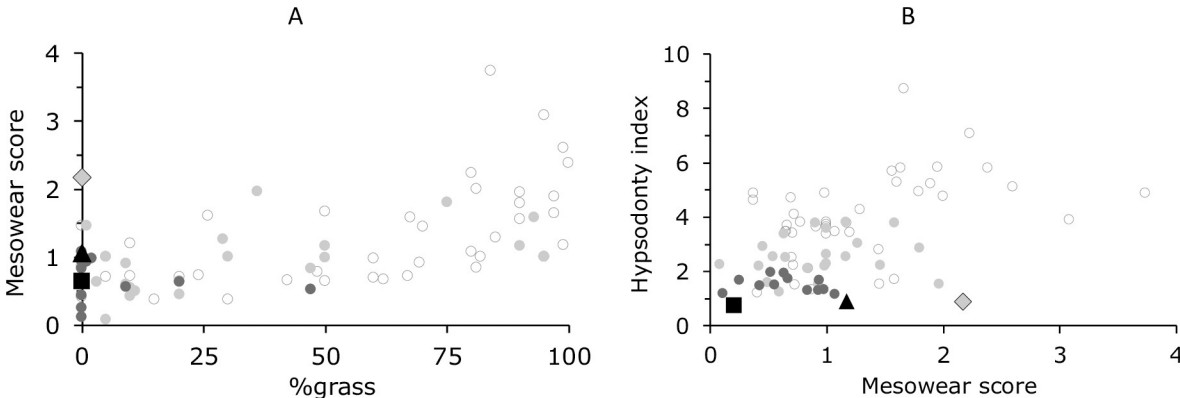

**Fig 24. Tapir mesowear compared to other ungulates.** Comparison (**A**) between diet and mesowear score and (**B**) of hypsodonty index and mesowear score distinguished between habitats. Circles = Data from Kaiser et al. [38]; Triangle = *T. terrestris*; Diamond = *T. bairdii*; Rectangle = *T. indicus*; white = open habitat; light grey = mixed habitat; dark grey & black = closed habitat. Hypsodonty index and habitat classification for tapirs from [3].

Fortelius [51], Taylor et al. [35] and many others [reviewed in 37]. We tested applying meso-wear analysis on tapirs, observing stable signals on the buccal tooth parts for low wear stages (Figs 19 and 20). Therefore, teeth worn to a high degree (macrowear score 4 and 5) were excluded to calculate the average mesowear score (Table 3). On the lingual side, mesowear sig-nals correlated with macrowear and were therefore assumed not to be robust (Figs 19 and 20), but the buccal side's mesowear scores appeared resistant to wear effects in the early wear stages. In principle, this finding supports the use of mesowear in tapirs, as e.g. applied by [11] for Lophialetidae, and extinct group of tapiroids.

We compared the mesowear score of the second molar of tapirs with seventeen species of strictly browsing ungulates from Kaiser et al. [38], at less than 5% grass in their natural diet. These had a mesowear score range between 0.08 (*Diceros bicornis*) and 1.46 (*Cephalophus syl-vicultor*). Calculating the mesowear scores of the $M^2$ of our tapirs (only specimens at macro-wear stages 1–3) in the same manner, *T. bairdii*, at 2.17, is the only species not within that range, due to its low occlusal relief (score 'low' = 83%, n = 6). Note that one would not expect such a low occlusal relief based on the bilophodont occlusion of tapirs–on the contrary. By contrast, the mesowear scores of *T. terrestris* (1.17) and *T. indicus* (0.20) were well within that range; for these tapir species the rate of low occlusal relief was less (31% in *T. terrestris*, n = 29; 0% in *T. indicus*, n = 5). No blunt cusps were present in macrowear stages 1–3 in any tapir spe-cies, and the portion of round cusps was more or less similar to the South American species (*T. terrestris* = 55%; *T. bairdii* = 50%) and distinctively less in *T. indicus* (20%). When taking the variable 'habitat' into consideration, *T. indicus* is the only species positioned as expected, whereas *T. terrestris* shows a rather high mesowear score for a browsing species in a closed habitat, and *T. bairdii* reaches a mesowear score comparable to grazing species (Fig 24). Com-paring mesowear score and hypsodonty, tapirs are the most brachydont species in this sample, yet *T. bairdii* reaches a mesowear score which is usually seen in more hypsodont species (Fig 24). The outlying mesowear score of *T. bairdii* might be a sign that mesowear scoring in tapirs is indeed compromised. As the ideal sample size for mesowear analysis (10–30) is not reached in *T. bairdii* and *T. indicus*, the results for these species should not be considered reliable [2]. *T. terrestris* was the only species with representative sample size (n = 29). An explanation for the rather high mesowear score in *T. terrestris* might be that in a freshly erupted cheek tooth of tapirs, the cusp shape is round. In mesowear scoring, unworn teeth are typically excluded [2, 38]. Restricting ourselves to $M^2$ of macrowear score 2–3, i.e. teeth with some wear, the tapir

mesowear scores remain basically similar: *T. terrestris* = 1.21 (± 1.22, n = 24); *T. bairdii* = 2.00 (± 1.22, n = 4); *T. indicus* = 0.20 (± 0.40, n = 5). This is explainable by the fact that also cusp from worn teeth rarely achieve a sharp stage (Fig 20), as the cusp is only sharp in the short stage between the beginning of wear and the development of a dental basin, where a sharpened facet of the enamel is present. On the buccal part, this case is rarely seen, as the buccal cusps are mostly not occluding. Whether the higher mesowear scores in *T. terrestris* and *T. bairdii* can be linked to the ingestion of fruits with hard seeds appears dubious to us, as we are not aware of reports of seed-cracking chewing behaviour in tapirs, and because the senior author observed intact large seeds in the stomach content and faeces of free-ranging *T. terrestris* investigated for a digestion study [52]. However, we note that the effect of large seeds on mesowear has not been investigated experimentally so far.

If one wanted to actually fit the tapir into a general, taxon-free mesowear scheme, one would have to adjust the numerical scores given to tapirs, so that their comparative scores were clearly among other browsing ungulates. However, it is evident that such an exercise would have little heuristic value. To date, there is no possibility to test whether a grazing feeding style would lead to a different mesowear score at similar macrowear stages in tapirs. Hence, it would require a (fossil) tapiroid for which, by other measures, a grazing diet could be confirmed, to gauge how such a feeding style would affect the mesowear score. Until such a possibility exists, one can just use the presence of tapirs in a dataset as representing the browsing niche, without the deviation of a mesowear score. Alternatively, if one assumed that the differences found at low *n* in the present study between the tapir species can be corroborated, one might attempt to compare the abrasiveness of these species' natural diets to understand the difference in mesowear.

### Comparisons natural habitat—zoo

To quantify wear progress between populations, we used the wear, in terms of RLD, of $M^3$ as a reference, as it is the latest erupting tooth, and compared the wear of premolars and molars between captive and free-ranging individuals in relation to that basis. This comparison allowed us to indirectly quantify tooth wear over time, assuming time span between eruption of a specific cheek tooth and $M^3$ to be constant between individuals. The mean tooth wear of zoo tapirs progresses slower than in their free-ranging conspecifics (Fig 21). This finding is quite unusual compared to other browsing herbivores. In the study of Kaiser et al. [28], zoo browsers mainly had a higher mesowear score (i.e., more wear) than their free-ranging conspecifics. Taylor et al. [29] came to the same conclusion for Rhinoceroses, as the mesowear score of the browsing *Diceros bicornis* showed more abrasion in zoo individuals, while in the grazing *Ceratotherium simum*, mesowear analysis showed less abrasion in captivity. Regarding mesowear score in *T. terrestris*, which was the only species reaching a representative sample size, captive lowland tapirs had a lower mesowear score (0.89 ±0.87, n = 18) than their free-ranging conspecifics (1.17 ± 1.12, n = 29). This finding is evident in every tooth position of *T. terrestris* (Table 3). Regarding *T. indicus*, the differences of average mesowear between zoo and free-ranging individuals is not consistent across all cheek teeth.

Heavy tooth wear can have a serious impact on animal health. For example in giraffes (*Giraffa camelopardialis*), mesowear classified captive individuals as grazing herbivores, which even lead to the assumption that this intensive tooth wear might have a negative impact on their longevity [27, 53]. In zoo tapirs, tooth and apical abscesses and teeth abnormalities have been reported [54, 55], but to our knowledge, no reports about unnatural excessively worn teeth exist. Furthermore, in captive Malayan Tapirs, a prevalence for resorptive tooth root lesions of 52% is described, which was much higher than in their free-ranging conspecifics

(6.5%) [56]. Therefore, regarding dental health in zoo tapirs, tooth wear seems not to be an issue, while other possible dental problems require more attention.

The most commonly assumed reason for increased tooth wear in captive browsers is a higher abrasiveness in their nutrition, compared to their natural diet [27–29]. Regarding feeding guidelines for tapirs, nowadays, the main portion of the diet should include mostly forage (most likely alfalfa hay) and also high fiber herbivore pellets [22, 57, 58]. Rose and Roffe [26], on the other hand, analyzed feeding strategies of Malayan Tapirs in different Zoos and showed that in only 2 out of 9 zoos the portion of forage achieved more than 50%, while no forage at all was fed in 5 out of 9 zoos. Given that tapirs often refuse to ingest grass hay [59], the forage given to tapirs–alfalfa hay or browse–should be of a comparatively low abrasiveness. When compared to non-forage items such as fruit, such forages might be nevertheless considered more contributive to tooth wear. Hence, we interpret the comparatively low level of wear in zoo tapirs as a sign that they receive less forage than their free-ranging counterparts would ingest. Thus, there appears to be no pressing issue regarding particular tooth wear in zoo tapirs. Clauss et al. [60] assumed the population of captive tapirs to be rather obese, due to increased energy intake and a lack of forage in tapir diets. Therefore, regarding zoo tapirs, we suggest following current feeding recommendations, including a higher portion of forage than currently present in many zoos.

## Conclusion

The–for perissodactyls–unusual orthal masticatory movement in tapirs, paired with anisodonty and their general tooth morphology, limits the application of the mesowear method on tapir teeth. We assume other parameters (macrowear/RLD) to be more robust to quantify wear in tapirs, and further conclude that zoo tapirs have less worn teeth than their free ranging conspecifics, which is unusual for browsing species but explainable by their zoo diets. Dental wear itself is therefore not assumed to cause health issues in the *ex situ* management of tapirs.

## Supporting information

**S1 Video. Chewing *Tapirus terrestris*.**
(MP4)

**S2 Video. Chewing *Tapirus terrestris*.**
(MP4)

**S3 Video. Chewing *Equus caballus*.**
(MP4)

**S4 Video. Chewing *Rhinoceros unicornis*.**
(MP4)

**S1 Dataset. Original dataset.**
(XLSX)

## Acknowledgments

We thank the participating museums for permitting the taking of dental imprints, and in particular Loic Costeur (Natural History Museum Basle) and Martina Schenkel (Zoological Museum of Zurich) for additional access to tapir skulls, and the Zurich Zoo (especially, Basil von Ah) for facilitating filming of chewing tapirs. We thank Michelle Aimée Oesch for the

photographs of skulls and dentition. Reviewers Mikael Fortelius and Joshua Xavier Samuels provided valuable comments that improved the original manuscript.

## Author Contributions

**Conceptualization:** Clemens J. M. Hohl, Marcus Clauss.

**Data curation:** Clemens J. M. Hohl, Thomas M. Kaiser, Marcus Clauss.

**Formal analysis:** Clemens J. M. Hohl, Daryl Codron, Marcus Clauss.

**Funding acquisition:** Jean-Michel Hatt.

**Investigation:** Clemens J. M. Hohl, Louise F. Martin, Dennis W. H. Müller, Marcus Clauss.

**Methodology:** Clemens J. M. Hohl, Thomas M. Kaiser, Marcus Clauss.

**Project administration:** Marcus Clauss.

**Resources:** Thomas M. Kaiser, Jean-Michel Hatt.

**Supervision:** Louise F. Martin, Marcus Clauss.

**Visualization:** Clemens J. M. Hohl, Marcus Clauss.

**Writing – original draft:** Clemens J. M. Hohl, Marcus Clauss.

**Writing – review & editing:** Daryl Codron, Thomas M. Kaiser, Louise F. Martin, Dennis W. H. Müller, Jean-Michel Hatt, Marcus Clauss.

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
