## [Decision Letter · Decision Letter 0]

19 May 2020

PONE-D-20-05755

Chewing, dental morphology and wear in tapirs (Tapirus spp.) and a comparison of free-ranging and captive specimens

PLOS ONE

Dear Prof. Clauss,

Thank you for submitting your manuscript to PLOS ONE. This is overall an excellent manuscript. Both reviewers had favorable evaluationsbut did have some suggestions for minor revision that need to be considered. Therefore, we invite you to submit a revised version of the manuscript that addresses the points raised during the review process.

We would appreciate receiving your revised manuscript by Jul 03 2020 11:59PM. To enhance the reproducibility of your results, we recommend that if applicable you deposit your laboratory protocols in protocols.io, where a protocol can be assigned its own identifier (DOI) such that it can be cited independently in the future. For instructions see: http://journals.plos.org/plosone/s/submission-guidelines#loc-laboratory-protocols

We look forward to receiving your revised manuscript.

Kind regards,

Matthew C. Mihlbachler, Ph.D.

Academic Editor

PLOS ONE

Journal Requirements:

This study was part of a project funded by the Swiss National Science Foundation (31003A_163300/1) to JMH. DWHM was financed by a Can-doc Forschungskredit (55220702) by the University of Zurich.

This study was part of project 31003A 163300/1 funded by theSwiss National Science

Foundation.

Reviewers' comments:

Reviewer's Responses to Questions

**Comments to the Author**

1. Is the manuscript technically sound, and do the data support the conclusions?

Reviewer #1: Yes

Reviewer #2: Yes

2. Has the statistical analysis been performed appropriately and rigorously? 

Reviewer #1: Yes

Reviewer #2: Yes

3. Have the authors made all data underlying the findings in their manuscript fully available?

Reviewer #1: Yes

Reviewer #2: Yes

4. Is the manuscript presented in an intelligible fashion and written in standard English?

Reviewer #1: Yes

Reviewer #2: Yes

5. Review Comments to the Author

Reviewer #1: This is a very interesting and useful piece of basic research with an additional applied aspect that is unlikely to be duplicated any time soon. The work is meticulously executed and documented, the results are clearly presented and the discussion is relevant and lucid. I have only one comment that may, or may not, require minor revision.

Comment: The mesowear method introduced by Fortelius & Solounias was not intended for dental morphologies lacking a distinct buccal wall on the upper teeth. In applying it to tapirs, it might be useful to add a description of how the scoring was done and a discussion of the influence of the bilophodont occlusion on the scores. I suspect that the unusually high scores might reflect some such a circumstance.

Reviewer #2: Overall, I enjoyed reading the manuscript and found it to generally be well done and clearly written. The authors have provided a detailed examination of chewing, tooth anatomy, and tooth wear in tapirs. The findings of the study highlight the orthal movements of the jaws when chewing, which is distinct from other perissodactyls. As a consequence of those movements, the teeth of tapirs have characteristic wear patterns as well. The authors have also compared wear patterns in samples of both wild caught and zoo specimens, revealing generally higher wear in wild populations.

Detailed below I have provided some suggestions for improving the manuscript, along with some points to consider. My main concerns are with the figures, which I feel could be improved in a number of ways that would make the paper easier to read.

Methods:

The Supporting Information does provide all of the raw data for the study, but that is not referenced anywhere in the manuscript text. Stating that specimens come from “different European museums” (Page 11, lines 94-95) in the text is not sufficient, particularly when the supporting table just uses museum acronyms. At minimum the authors should list the museums visited (and acronyms) in the Methods, and then provide a reference to the supporting table listing the complete sample details.

Page 12, line 117 to Page 13, line 150 and Figure 1 present quantitative measurements taken for this study. While the text descriptions and figure are relatively clear, I think much of this would be well suited to present differently. Presentation of measurements in a table, listing abbreviations and definitions, will make them clear and easy for readers to locate. Figure 1 not only shows measurements themselves, but also a series of relative measures (ratios) derived from them. Restricting the figure to presenting only the actual measurements and showing measurement definitions (both direct and relative measures) in a Table would probably make this all more easily interpretable.

Page 13, lines 154-155 seem to be just randomly placed, rather than flowing directly from the preceding text (“Quantitative measurements”). I would recommend relocating this earlier in the methods, following some of the other basic information regarding the dental formula of tapirs (Page 12, line 114).

The Methods on Page 13, line 158 to Page 15, line 191, Results Page 23, lines 420-421, and Table 2 describe derivation of macrowear and mesowear scores. Here, the authors have developed a scoring system for wear stages of individual teeth. A master’s thesis by Gibson (2011) looked at the population structure of a large sample of fossil tapirs, and developed a method for roughly aging tapirs into 7 categories based on tooth eruption and wear stages. Comparison of those classes to the macrowear based age classes developed here would be interesting.

Gibson, M.L., 2011. Population Structure Based on Age-Class Distribution of Tapirus polkensis from the Gray Fossil Site Tennessee. MS Thesis. East Tennessee State University.

Results:

The results are dominated by quite a few Figures, which are each described as presenting correlations between variables studied. Rather than a large number of bivariate plots in the text, I wonder if this content could be presented in a more straightforward fashion using tables, and then potentially providing plots in supporting material. For example, I think Figures 13-15, and 17 could all be presented as tables.

Figures 4, 6-10, 12-14, etc. would be more readable with some enlargement of fonts.

Also, in Figure 4 there are two distinct gradients used to indicate tooth position in A and B. I find the direct transition of tones in Fig. 4B to be more easily interpreted, and suggest changing A to be a similar sort of transition in grayscale tones.

Figure 5 would be more appropriately included earlier in the Methods (lines 109-112 would work), and can be referred to in the Results as well.

Figures 6 through 10 include a line, but that is not defined anywhere. I assume it is just showing equal proportions, but it should be clear that this is not some sort of regression line or something else.

Discussion:

Page 28, line 508 to page 29, line 510 discusses the lophodont pattern and inferred orthal movement of the tapirs. Many mammals with some form of lophodont occlusal morphology (including loxodont, ptychodont, and selenolophodont teeth) have primary orientation of those folds/ridges perpendicular to the primary direction of jaw movement (Ungar 2010). Proboscideans (loxodont), rodents (many lophodont or ptychodont lineages), and some marsupials (like kangaroos and wombats) show mediolaterally oriented lophs on their teeth and a primary orthal (propalinal) movement of the jaws when chewing. Mentioning those sorts of common patterns helps support the interpretations here.

Ungar, P.S., 2010. Mammal teeth: origin, evolution, and diversity. JHU Press.

Page 29, line 531 to page 30, 537 discusses the size of the temporalis muscle and sagittal crest. Outside of ungulates, there are many other herbivorous mammals with large and prominent temporalis and sagittal crest, including many rodents like beavers and chisel tooth digging burrowers (Samuels 2009) and primates like Gorillas.

Figure 23 caption, the word comparison is redundant at the start of the sentence.

Page 34, lines 647-654 discuss determination of tapir age by wear, Gibson’s (2011) study of tooth eruption and wear mentioned above would be directly relevant to this section.

Page 35, line 678 to Page 37, line 724 discuss the mesowear of tapirs, and the fact Tapirus bairdii has a greater mesowear than is typical of other brachydont species. Rather than the methodology being compromised, as mentioned in lines 694-695, it is important to remember mesowear reflects attrition of the tooth over a substantial portion of the animal’s life (Fortelius and Solounias 2000, Davis and Pineda-Munoz 2016), and thus could potentially be a consequence of multiple factors. What the authors mention in lines 721 to 724 is what I suspect is the most important part of this and should be expanded upon. While grazing or feeding on grit-covered plants in relatively arid habitats may be the typical way teeth wear in herbivorous mammals, I suspect consumption of fruits and nuts with hard seeds or pits could also potentially lead to higher attrition of the teeth in wild tapir populations. Though such foods might not be the bulk of the diet in a population, they could still have a substantial impact on tooth wear. Additionally, the actual ages of individuals in these relatively small samples is unknown. If the T. bairdii sample had individuals of more advanced age than the other two tapir species (or other previously published brachydont browsers), they would be expected to have relatively more advanced wear.

6. PLOS authors have the option to publish the peer review history of their article (what does this mean?). If published, this will include your full peer review and any attached files.

Reviewer #1: Yes: Mikael Fortelius

Reviewer #2: Yes: Joshua Xavier Samuels

---

## [Author Response · Author response to Decision Letter 0]

22 May 2020

Please see the detailed reply letter.

---

## [Editor Report · Decision Letter 1]

3 Jun 2020

Chewing, dental morphology and wear in tapirs (Tapirus spp.) and a comparison of free-ranging and captive specimens

PONE-D-20-05755R1

Dear Dr. Clauss,

We’re pleased to inform you that your manuscript has been judged scientifically suitable for publication and will be formally accepted for publication once it meets all outstanding technical requirements.

Kind regards,

Matthew C. Mihlbachler, Ph.D.

Academic Editor

PLOS ONE
---

## [Editor Report · Acceptance letter]

5 Jun 2020

PONE-D-20-05755R1 

Chewing, dental morphology and wear in tapirs (*Tapirus* spp.) and a comparison of free-ranging and captive specimens 

Dear Dr. Clauss:

I'm pleased to inform you that your manuscript has been deemed suitable for publication in PLOS ONE. Congratulations! Your manuscript is now with our production department. 

Kind regards, 

on behalf of

Dr. Matthew C. Mihlbachler 

Academic Editor

PLOS ONE